# GenQA: Generating Millions of Instructions from a Handful of Prompts

## Abstract

Most public instruction finetuning datasets are relatively small compared to the closed source datasets used to train industry models. To study questions about finetuning at scale, such as curricula and learning rate cooldown schedules, there is a need for industrial-scale datasets. However, this scale necessitates a data generation process that is almost entirely automated. In this work, we study methods for generating large instruction datasets from a single prompt. With little human oversight, we get LLMs to write diverse sets of instruction examples ranging from simple completion tasks to complex multi-turn dialogs across a variety of subject areas. When finetuning a Llama-3 8B base model, our dataset meets or exceeds both WizardLM and Ultrachat on both knowledge-intensive leaderboard tasks as well as conversational evaluations. We release our dataset, the "generator" prompts that created it, and our finetuned model checkpoints.

## 1 Introduction

Datasets for language model finetuning are typically crafted by hand, crowdsourced from a pool of human annotators, or built by prompting other large language models to edit and augment existing human written datasets (Ding et al., 2023; Köpf et al., 2024; Mukherjee et al., 2023; ShareGPT, 2023; Xu et al., 2023) The requirement of human inputs makes dataset curation an arduous and expensive process. These costs have also resulted in a split between academic and industrial finetuning practices, with academic datasets comprising hundreds or thousands of samples, and industrial datasets comprising tens of millions.

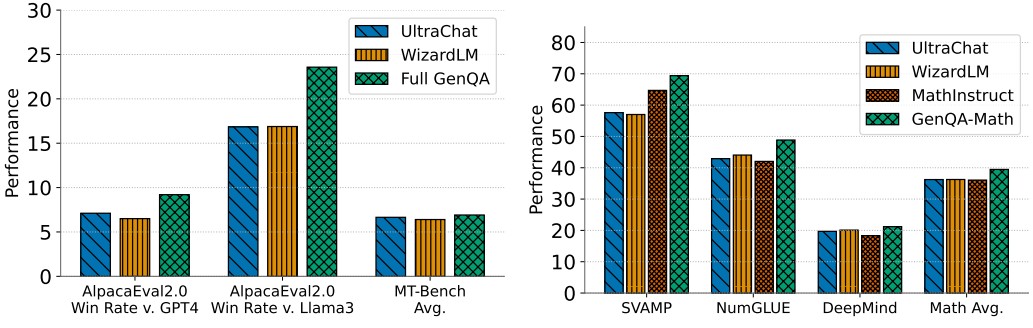

Figure 1: (left) Performance of llama-3-8b finetuned on GenQA for instruction following benchmarks (see Table 2). (right) Finetuning on the math split of GenQA demonstrates strong mathematical task performance (see Table 4).

We introduce GenQA, an instruction dataset that is written autonomously by an LLM without being conditioned on human-written questions. We demonstrate that a single hand-written meta-prompt can be used to extract millions of diverse questions from an LLM. Surprisingly, this automated process can result in high quality datasets that compete with (or even surpass) datasets created using extensive human labor.

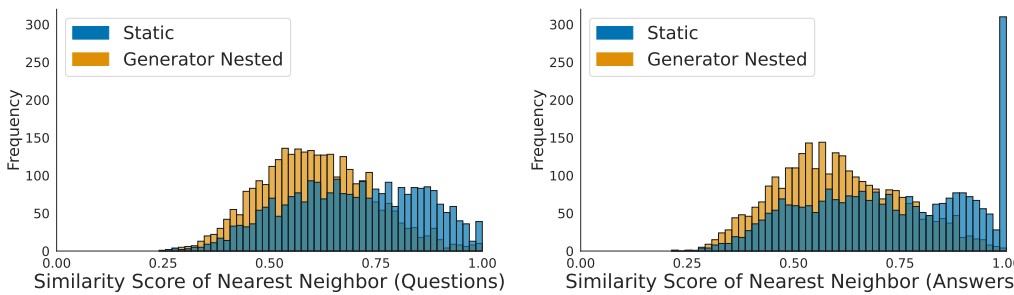

Figure 2: Compared to static prompts that simply ask an LLM for a question/answer pair, our generator prompting strategy results in much higher diversity and more unique questions. Lower similarity scores indicate more diversity, see Section 3.3.

Automated dataset creation may seem challenging, as most language models suffer from a lack of randomness. When asked many times to write an instruction/response pair, LLMs may produce a low diversity dataset with many duplicate questions. To extract diverse questions from an LLM in an automated way, we propose *generator prompts,* a prompting strategy that boosts the randomness of LLM outputs. By using a small number of generator prompts, we are able to create a large instruction dataset containing many different question styles across a wide array of subjects.

The GenQA dataset has a few key attributes that make it useful for research purposes. First, the GenQA dataset natively contains several different splits produced using different kinds of meta prompts. Second, the GenQA dataset is fairly large, and is meant to reflect the scale of the more than 10M instruction samples reportedly used to fine tune Llama 3. Finally, our empirical evaluation suggests that models trained on GenQA perform well (Figure 1). In addition to demonstrating the ease with which dataset creation can be automated, we hope the combination of diversity and scale in the GenQA dataset will enable open research on industrial-scale finetuning practices.

## 2 RELATED WORK

One of the earliest forms of modern data for language model finetuning was the Flan collection (Wei et al., 2021). The multi-task dataset used to train the T0 model family (Sanh et al., 2021) as well as the work of Chung et al. (2022) and Xu et al. (2022) expanded these data to include thousands of tasks, achieving further improvements over the original Flan. These datasets, while large and mostly factually correct, suffer from grammatical errors and other major text quality issues. Ouyang et al. (2022) combined multi-task data and a reinforcement learning objective to produce InstructGPT, a finetune of GPT-3 with improved controllability and utility for downstream users.

Next, Wang et al. (2022) showed that an existing finetuned model such as InstructGPT (Text-Davinci-003) could be used to produce instruction-output pairs that were in turn useful to finetune other foundation models in an output based distillation process. One of the most notable demonstrations of this technique was the landmark work that turned the base Llama model (Touvron et al., 2023) into a highly capable instruction-following version, Alpaca, using a dataset of just 52k outputs sampled from Text-Davinci-003 (Taori et al., 2023). As distilled datasets became more popular, the community began constructing others by either filtering and/or augmenting existing datasets in specific ways. Xu et al. (2023) augmented the original Alpaca dataset (Taori et al., 2023) to produce *Evol-Instruct*, sometimes referred to as *WizardLM*, using a pipeline that "evolves" existing instructions with five types of meta prompts that constrain, deepen, concretize, increase reasoning steps, and generally complicate the original input.

These model augmented instruction datasets were very small, often containing only tens of thousands of examples. To increase the scale at which open source models could be instruction tuned, Mukherjee et al. (2023) sampled the dataset proposed by Chung et al. (2022) and rewrote the responses using ChatGPT and GPT-4 (Brown et al., 2020; Achiam et al., 2023). In another approach, Teknium (2023) and Wang et al. (2023) methodically combined multiple sources of existing instruction tuning datasets to create very large singular datasets. For the domain of mathematics in particular, Yue

| Split | Questions | Description |
|---|---|---|
| Academic | 4,210,076 | QA on a range of academic topics |
| MMLU | 2,409,841 | QA on the topics found in the MMLU dataset |
| Multiple Choice | 372,610 | Multiple choice questions on diverse topics |
| Writing | 932,362 / 1,864,724 | Compositional writing and editing of documents |
| Task | 1,004,179 / 1,515,280 | Non-compositional text-based tasks |
| Code | 513,483 | QA about programming topics in various languages |
| Math | 515,509 / 1,104,324 | Math questions, elementary to graduate level |
| Dialog | 819,154 / 3,222,818 | Multi-turn conversations containing explanations or advice |
| General | 304,920 | QA about pop culture and daily life |

Table 1: GenQA contains 11,082,134 questions (15,518,076 counting each conversation turn separately) broken into nine splits, each of which were produced using different prompts. In total, the dataset contains approximately 2.8 billion whitespace delimited words.

et al. (2023) compiled many existing mathematical reasoning datasets into a single compendium and further supplemented them using GPT-4 (Achiam et al., 2023) to produce MAmmoTH. However, all of these approaches were hampered by the fact that machine generated data tends to lack diversity and concentrate around few modes (Zhang et al., 2024).

In a concurrent work, Xu et al. (2024) extract instructions from Llama3 models by prompting them with an empty string, which often results in a random instruction. This same strategy was used to create the *general* split of GenQA using GPT-3.5, although the GPT models produce a random answer rather than a random question (see Appendix A.2). Our initial experiments with instruction extraction found the empty string approach to be ineffective with commercial models; After making 2M queries to GPT-3.5, the empty string strategy resulted in only 304K unique answers (15% uniqueness). We found higher diversity and better control over question topics using our generator prompt strategy, which yielded 85% unique answers in all of our experiments.

Our approach for building a completely machine generated dataset differs from these prior works both in its scale ($> 10M$ samples) and in our use of specially constructed prompts that result in a set of generated instructions and responses with high diversity in both their structure and the topics that they cover.

## 3 THE GENQA DATASET

Table 1 lists the splits of GenQA, along with the number of instructions in each and a brief description of the split. The number of tokens in questions and responses is shown in Figure 3. See Appendix A.5 for more detailed token counts of multi-turn conversations. In the main paper below, we describe the methodology used to create the dataset, and give concrete examples of meta-prompts used to create the Academic split. Appendix A.2 lists the generator meta-prompt used to create every split, along with a representative question and answer from each split. All questions were created by Gemini Pro 1.0, with the exception of the General split, which also involved GPT-3.5 (see Appendix A.2 for more details).

In the following sections, we explain how our specially crafted *generator prompts* work, and how we used them to construct the GenQA dataset. We then perform a scientific analysis of the dataset to understand which prompting strategies were most effective, and how generator prompts should be designed to promote diversity.

### 3.1 GENERATOR PROMPTS

The quality and diversity of training data are crucial to instruction tuning. Unfortunately, it can be difficult to induce an LLM to produce a large amount of diverse content. Here, we introduce

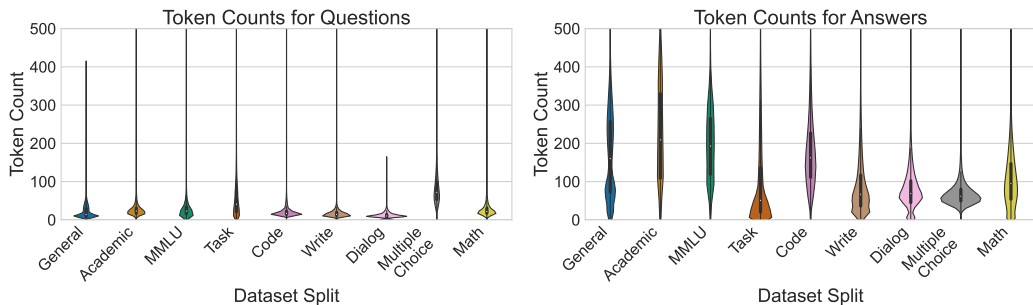

Figure 3: White-space token count for questions (left) and answers (right). Charts are truncated at 500 tokens. Some answers contain over 6K tokens, see Appendix A.5 for the full tail of the distribution.

*generator prompts.* We then study the best way to formulate these prompts by quantifying the level of data diversity they yield.

A naive method for automating content generation is simply to choose a topic, and then construct a *static prompt* that asks for content on that topic. Consider the toy example of generating a long list of colors. This can be done by feeding the following static prompt to Gemini 1.0 Pro many times:

```
State a random color.  Don't output anything but the color.
```

The use of a static prompt yields low diversity. Running the above prompt through the Gemini Pro 1.0 language model 1000 times produces only 33 unique outputs.

A *generator prompt* boosts diversity by asking a model to produce a long list of possible choices, and then select one of the candidates at random. For example, consider the following prompt:

```
First, print the heading "Colors:", followed by a numbered list of
100 colors.  Then, print the heading "Chosen color:".  Then print
color number {N} on a line by itself.
```

The placeholder N should be replaced with a random number each time the prompt is invoked. When run 1000 times, this prompt yields 383 different colors. One can also use two nested generators as follows.

```
First, print the heading "Colors:", followed by a numbered list
of 100 different colors.  Then, print the heading "Chosen color:".
Then print color number {N1} on a line by itself.  Then, print
the heading "Color variants:".  Then print a numbered list of 100
different color variants that look like color number {N1}, and
don't appear on the original "Colors:" list.  Then, print the
heading "Chosen variant:".  Then print variant number {N2} on a
line by itself.
```

This prompt yields 782 unique colors from 1000 runs.

We hypothesize that the output diversity produced when using a *generator prompt* comes from several sources. First, by explicitly creating a list, in the above example we guarantee that there will always be at least 100 candidates, and that these candidates are unique (assuming the LLM followed the provided directions). Second, the process of creating the list requires many sequential samples to be drawn from the model. If the temperature is warm, this compounded randomness makes it highly unlikely that the same list will be produced more than once.

## 3.2 A STUDY OF GENERATOR PROMPTS FOR DATASET CREATION

Here, we study how to apply the idea of generator prompts to create random question/answer pairs. We consider several different prompt types in order of increasing complexity. Then, we rigorously study the level of diversity these prompts produce.

Initial attempts to create the Academic split using static prompts resulted in low diversity. For example, one could choose this prompt:

**Static:** `Write a random complex question and its long answer. Begin your question with "Question:" and your answer with "Answer:`

This results in many repeated/identical outputs, motivating us to provide additional context to our prompt. We consider the following prompt conditioned on `random_topic`, which is randomly selected from a list of topics written ahead of time by Gemini (see Appendix A.6 for full list).

**Static-Conditional:** `Write a complex question from the domain of {random_topic}. Then write the long answer. Your question should not contain the words "{random_topic}". Begin your question with "Question:" and your answer with "Answer:"`

To further boost randomness and prevent the model from collapsing into a single mode around each topic, we consider the following generator prompt, also conditioned on `random_topic`.

**Generator-Conditional:** `List 40 subtopics in the domain of {random_topic}. State subtopic {N}. Then write a question that is not about subtopic {N}, but can only be answered with expertise in subtopic {N}, and then write the answer. Both the question and answer should be long. The name of the subtopic should not appear in the question. Begin your questions with "Question:" and your answer with "Answer:". Be creative.`

Conditioning on a topic prevents the model from collapsing into a small number of modes, but it also constrains the range of possible topics. In the example shown in **??**, we see that one can produce randomness using the nested generator approach. Applying this idea results in the following:

**Generator-Nested:** `List 60 topics that you can answer questions about. State topic {N1}. Then write 60 subtopics about topic {N1}. Then state the subtopic {N2}. Then write a question that is not about subtopic {N2}, but can only be answered with expertise in subtopic {N2}. Then write the answer. Both the question and answer should be long. The name of the subtopic {N2} should not appear in the question, and none of the words in subtopic {N2} should be reused in the question. Begin your questions with "Question:" and your answer with "Answer:". Be creative.`

This method has a potential drawback: The LLM sees the selected indices before writing the list, and this may influence the order of the listed items. For this reason, we consider the following construct:

**Generator-Uniform:** `List 60 topics that you can answer questions about. Choose a topic uniformly from this list, and state it. Then write 60 subtopics about the chosen topic. Then choose a subtopic uniformly from this list, and state it. Then write a question that is not about the subtopic, but can only be answered with expertise in the subtopic. Then write the answer. Both the question and answer should be long. The name of the subtopic should not appear in the question, and none of the words in subtopic should be reused in the question. Begin your questions with "Question:" and your answer with "Answer:". Be creative.`

In this construction, the random index is not available to the LLM when the lists are being constructed, as the index it chosen via sampling rather than appearing in the prompt.

Remark: The Gemini model tends to interpret instructions quite literally. If we ask for a question about Cultural Anthropology, we are likely to get a question about Cultural Anthropology *per se*, such as "Who is the father of Cultural Anthropology" or "What is that most famous textbook in Cultural Anthropology." We avoid this caveat by prompting for a question that is "not about the subtopic, but can only be answered with expertise in the subtopic."

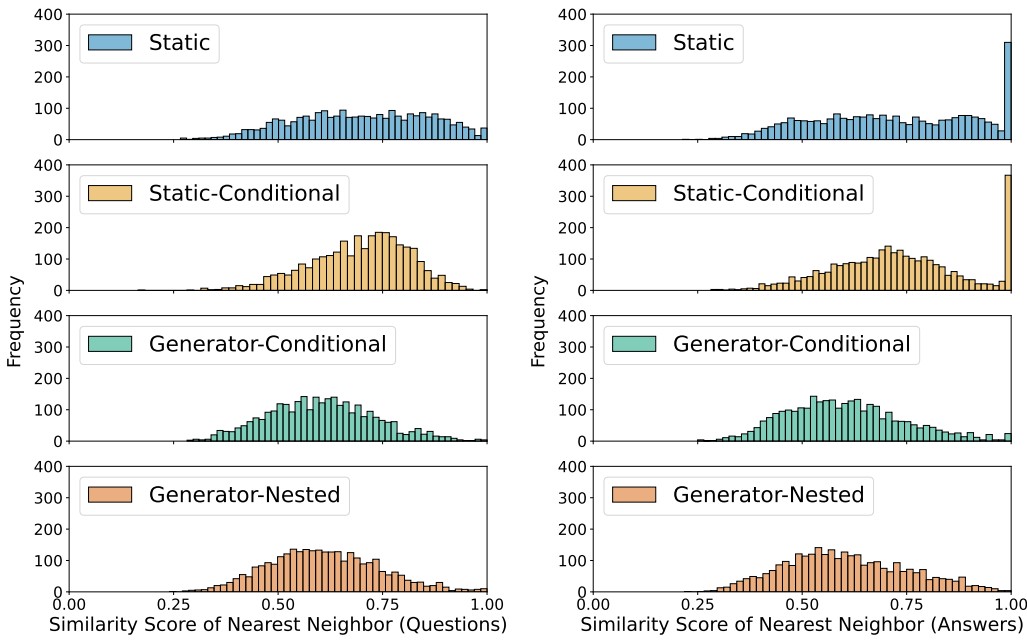

Figure 4: Comparison of nearest-neighbor similarity scores in the Academic split. The generator-conditional and generator-nested strategies perform best.

### 3.3 THE SCIENCE OF GENERATOR PROMPTS: WHICH CONSTRUCTION YIELDS THE MOST DIVERSITY?

To evaluate these prompting strategies, we generate a separate dataset of size 3000 for each prompt type.[1] To measure the level of diversity produced by each prompt, we used the all-MiniLM-L6-v2 retrieval model Wang et al. (2020) to encode the first 2 sentences of each question (or answer) into 384-dimensional dense vectors. For each question (or answer), we record its cosine similarity to its nearest-neighbor. A similarity score of 1.0 indicates that a question has an exact duplicate, while smaller values indicate that an input is more unique.

Figure 4 compares the similarity scores within each dataset, computed on either questions or answers. The static prompting strategy resulted in high similarity between nearest neighbors and a higher number of duplicates. The generator-conditional and generator-nested prompts, which were used to create the final academic split, yield the highest diversity. Please refer to the appendix figure in A.3 for the full analysis of the prompting strategies evaluated in the Academic split. Similar analysis is conducted for the Task split and can also be found in Appendix A.3.

### 3.4 ADDING A SUFFIX TO BOOST RANDOMNESS

The prompts above end with the phrase "Be creative." We call this a *randomness booster*. During the creation of the splits, we randomly append one of the following boosters to the end of the prompt every time it is invoked: "Be creative," "Be different," "Be smart," "Be weird," "Don't ask the first thing you think of," "Be creative and don't ask the first thing you think of," or an empty string (no booster).

To assess the effect of the booster, we sample an equal number of question and answer pairs generated with and without a booster from each split in our dataset ($n = 200$ for each type). Following the same procedure in Section 3.3 to analyze diversity, we demonstrate the impact of boosters for the Academic split in Figure 5. The presence of the booster improves diversity across Academic, Task, Multiple Choice, MMLU and Dialogue splits. We include further analysis of boosters in Appendix A.4.

---

[1]We did exact deduplication on the released the dataset, but this study was performed pre-deduplication.

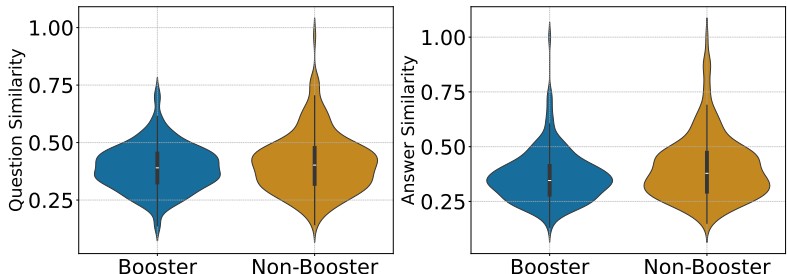

Figure 5: Applying a random booster to the prompts for the Academic split improves data diversity.

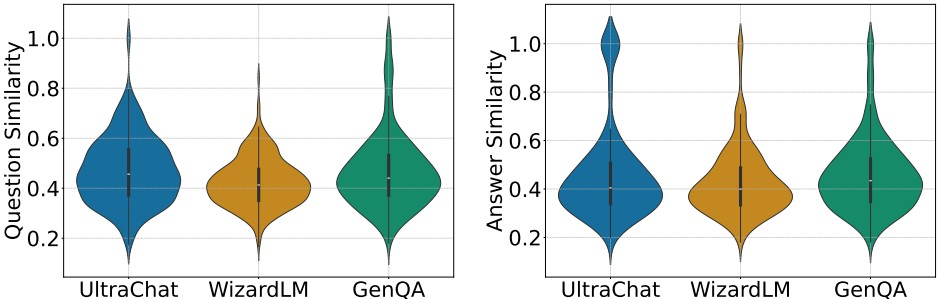

Figure 6: Comparison of nearest neighbor ($k = 1$) similarity across GenQA, UltraChat, and WizardLM.

## 3.5 ASSEMBLING THE FINAL DATASET

The GenQA dataset was created by constructing generator prompts, either topic-conditioned or nested, for each split. Prompts used for each split and example questions generated by the prompts are listed in detail in Appendix A.2. For each split, the generator prompts were fed through Gemini Pro 1.0 many times and the outputs were parsed into questions and answers. The final questions were deduplicated using an exact match criteria on the first two sentence of the questions.

## 3.6 COMPARING DIVERSITY IN OTHER FINETUNING DATASETS

We compare the diversity of GenQA to existing finetuning datasets *WizardLM* and *UltraChat* by uniformly sampling equal sized subsets from each dataset and analyzing their diversity in the same manner as Section 3.3. We observe that the diversity of GenQA is on-par with that of the reference datasets, as seen in Figure 6.

## 4 FINETUNING EXPERIMENTS

To demonstrate the quality of GenQA for language model finetuning, we perform an empirical evaluation against other strong finetuning datasets.

## 4.1 FINETUNING SETUP

We tune a Llama-3-8B (AI@Meta, 2024) model with the default chat template on GenQA and two existing instruction finetuning datasets: UltraChat (Ding et al., 2023) and WizardLM (Xu et al., 2023). To evaluate the final models we consider tasks from the Huggingface Open LLM Leaderboard and two instruction-following benchmarks. We use the Hugging Face Alignment Handbook codebase Tunstall et al. (2023) for our finetuning runs and the same set of standard hyperparameters for each dataset. Full details are provided in Appendix A.1. The baseline instruction datasets and evaluation benchmarks we consider are described below.

Table 2: Performance on two different benchmarks that measure a model's ability to engage in coherent, informative, and engaging conversations, AlpacaEval and MT-Bench.

| Dataset | AlpacaEval 2.0 | | | MT-bench | | |
|---|---|---|---|---|---|---|
| | v.s. GPT4 | v.s. Llama-3-8B-Instruct | Length | 1-round | 2-round | Avg. |
| WizardLM | 7.11 | 16.84 | 1522 | 7.18 | 6.12 | 6.65 |
| Subset GenQA (135M tokens) | 7.75 | 17.49 | 1498 | 7.26 | 6.06 | 6.67 |
| UltraChat | 6.50 | 16.88 | 1282 | 6.93 | 5.88 | 6.40 |
| Subset GenQA (238M tokens) | **9.31** | 20.41 | 1096 | 7.49 | 6.15 | 6.82 |
| Full GenQA | 9.20 | **23.57** | **1060** | **7.55** | **6.26** | **6.91** |

Table 3: Performance on various reasoning, knowledge, and truthfulness benchmarks. GenQA is on par with the other models, including Llama-3-8B-Instruct.

| | ARC_E | ARC_C | BoolQ | HellaSwag | MMLU | OpenBookQA | PIQA | TruthfulQA | Winogrande | Avg |
|---|---|---|---|---|---|---|---|---|---|---|
| Llama-3-8B | 52.99 | 77.86 | 80.95 | **79.15** | 62.01 | 34.80 | 80.79 | 43.80 | 73.56 | 65.10 |
| Llama-3-8B-Instruct | **56.91** | 79.67 | 83.15 | 75.79 | **63.81** | 42.80 | 78.73 | 51.70 | 71.67 | 67.14 |
| WizardLM | 55.12 | 79.63 | 82.02 | 78.38 | 61.42 | 45.40 | 80.79 | **53.03** | **74.59** | **67.82** |
| Subset GenQA (135M tokens) | 55.46 | **80.30** | 83.73 | 78.65 | 60.88 | **46.00** | 81.18 | 48.16 | 74.51 | 67.65 |
| UltraChat | 54.86 | 79.76 | 83.39 | 78.60 | 62.05 | 44.20 | **81.45** | 49.05 | 73.88 | 67.47 |
| Subset GenQA (238M tokens) | 55.38 | 80.22 | **83.82** | 78.51 | 60.51 | 45.60 | 81.18 | 48.26 | 74.11 | 67.51 |
| Full GenQA | 55.46 | 80.13 | 83.70 | 78.81 | 61.07 | **46.00** | 81.28 | 47.06 | 74.03 | 67.50 |

*WizardLM-Evol-Instruct-V2* (Xu et al., 2023) contains 196k single-turn instructions. The dataset is developed by starting with initial instructions from a base dataset such as Alpaca (Taori et al., 2023). It is then enhanced using a large language model like GPT-4, which incrementally increases the complexity of the instructions through various strategies[2].

*UltraChat* (Ding et al., 2023) is another synthetic instruction dataset, specifically focusing on multi-turn conversational abilities. The dataset is generated by simulating conversations between two large language models on three different topics: general questions, writing, and assistance, aiming to ensure diversity. In this paper, we use a filtered version of UltraChat with a total of 200k multi-turn instructions [3].

**Evaluation** To demonstrate the capabilities of our finetuned models, we evaluate them on a variety of general benchmark tasks and conversational benchmarks. For general benchmark tasks, we include ARC (Clark et al., 2018), BoolQ (Clark et al., 2019), HellaSwag (Zellers et al., 2019), MMLU (Hendrycks et al., 2020), OpenBookQA (Mihaylov et al., 2018), PIQA (Bisk et al., 2020), TruthfulQA (Lin et al., 2021), and Winogrande (Sakaguchi et al., 2019). This diverse range of benchmarks assesses the models' reasoning, knowledge, and truthfulness. Additionally, we report scores for AlpacaEval 2.0 length-control and MT-bench. These benchmarks test the models' conversational abilities and their capacity to follow instructions.

**Rebalancing the GenQA Splits** The GenQA dataset is not balanced across splits, with the Academic split comprising $\sim 38\%$ of the entire dataset in its raw form. We find that we get best finetuning performance using adjusted sampling ratios that up-weight the smaller splits. See Table 5 in Appendix A.1. We refer to the rebalanced version of the dataset as "Full GenQA" and the "Subset GenQA" versions at various token counts are a further random sample of the rebalanced dataset.

## 4.2 RESULTS OF FINETUNING

We showcase the results of finetuning on different datasets in Table 2. The model finetuned on GenQA achieves the highest scores on both AlpacaEval and MT-bench. Additionally, we report the

---

[2] https://huggingface.co/datasets/MaziyarPanahi/WizardLM_evol_instruct_V2_196k

[3] https://huggingface.co/datasets/HuggingFaceH4/ultrachat_200k

Table 4: Performance on various Math reasoning tasks with 5 shots Chain-of-Thought (CoT) prompting. The Math split of GenQA outperforms all other datasets including a similarly sized but random subset of GenQA, and a Math specific instruction tuning dataset, MathInstruct (Yue et al., 2023).

| | Math | GSM8K | SVAMP | NumGLUE | DeepMind | SimulEq | Avg. |
|---|---|---|---|---|---|---|---|
| WizardLM Subset GenQA (125M tokens) | 17.10 | **58.07** | 57.60 | 42.90 | 19.70 | **21.98** | 36.23 |
| | 18.68 | 56.56 | 65.00 | 46.64 | 19.90 | 19.26 | 37.67 |
| UltraChat Subset GenQA (238M tokens) | 18.06 | 57.85 | 57.00 | 44.05 | 20.10 | 20.43 | 36.25 |
| | 18.66 | 55.72 | 68.90 | 47.41 | 19.30 | 20.04 | 38.33 |
| MathInstruct (114M tokens) | 19.56 | 56.79 | 64.70 | 42.03 | 18.30 | 14.98 | 36.06 |
| GenQA-Math (222M tokens) | **20.30** | 56.86 | **69.40** | **48.85** | **21.20** | 20.23 | **39.47** |

results of Leaderboard tasks in Table 3. The model finetuned on GenQA achieves comparable results to those of models finetuned on the baseline datasets. Overall we find that GenQA is performant when evaluated from a token-for-token sample complexity perspective against other datasets, and we also observe even further improvement if we leverage its size by training on its totality.

**Token-for-Token**    To ensure a fair comparison with GenQA, given its significantly larger scale than the WizardLM and UltraChat datasets, we randomly sample a subset from GenQA to match the token count of the baseline datasets. In Table 2 and Table 3, we present the results of the model finetuned on a subset of GenQA, referred to as "Subset GenQA," which has the same number of tokens as the baseline dataset it is paired with.

The token-for-token comparison in Table 2 reveals that the model finetuned on GenQA outperforms the UltraChat dataset according to both instruction-following benchmarks evaluated and also is comparable to tuning on the WizardLM dataset. Measured on the series of knowledge and reasoning leaderboard benchmarks tabulated in Table 3, GenQA is also competitive. We find that measured across all benchmarks in aggregate, and controlling for token count, the average performance of GenQA slightly beats UltraChat and slightly underperforms WizardLM, but is generally comparable[4].

In summary, our token-for-token comparison indicates that despite being *totally machine generated* from a handful of prompts without conditioning on human written questions, GenQA yields high-quality finetuned models. The baseline datasets we compare it to are derivatives of existing data, requiring further calls to state of the art models like GPT-4 during their augmentation process, and/or must undergo additional data curation steps to enhance their complexity.

**Training on Individual Splits**    We also conduct a series of experiments where we finetune the model exclusively on the Math split of GenQA. This allows us to compare it to other large expert datasets like MathInstruct (Yue et al., 2023), which is specifically designed to enhance Mathematical reasoning. We evaluate the finetuned models on various Mathematical reasoning datasets including Math (Hendrycks et al., 2021), GSM8K (Cobbe et al., 2021), SVAMP (Patel et al., 2021), NumGLUE (Mishra et al., 2022b), DeepMind (Saxton et al., 2019), and SimulEq (Mishra et al., 2022a). As shown in Table 4, a model finetuned on the GenQA Math subset outperforms the baselines in all benchmarks except SimulEq, where the models finetuned on WizardLM and UltraChat perform marginally better.

**Full GenQA: Is bigger better?**    We observe that under the instruction-following evaluation, the model trained on all of GenQA performs better than the models trained only on WizardLM or UltraChat, or the subset of GenQA, in nearly all cases. Furthermore, on the Leaderboard tasks for which GenQA outperforms its peers, extending training from the subset to the full GenQA further improved scores.

---

[4]To calibrate expectations, we observe that finetuning on any of the instruction datasets achieves results comparable to the official Llama-3-8B-Instruct model, surpassing the base model Llama-3-8B handily.

It should be noted, however, that on Leaderboard tasks where the subset of GenQA did not yield better performance than the baselines, continued training was not able to make up the difference. This suggests that GenQA still has some blindspots, and like other popular datasets it may benefit from being mixed into a larger cocktail (Teknium, 2023).

## 5 CONCLUSION

GenQA is an instruction dataset written autonomously by an LLM without conditioning on human questions or using complex multi-stage pipelines. Beyond the obvious uses of the public GenQA samples to improve performance of open source models, we hope that the methods in this paper can serve as a Swiss army knife for easily creating datasets for other domains. Our experiments indicate that prompt engineering alone can yield millions of diverse training samples with quality as good as (or in some cases surpassing) high-cost human labelling. The generator prompt strategy can be used to quickly generate datasets anew, or to create data to cover the blindspots of other existing data sources.

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

# A APPENDIX

## A.1 FINETUNING HYPERPARAMETERS

We utilize the Hugging Face Alignment Handbook codebase Tunstall et al. (2023) for our finetuning runs and the same set of standard hyperparameters for each dataset. During finetuning, we employ AdamW (Loshchilov & Hutter, 2017) without weight decay. We warm up for the first $10\%$ of the total steps, and after the warm-up period, we use the learning rate to 1e-6 and utilize a cosine annealing scheduler, reducing it to $0$. Our setup includes a batch size of 8 per device, an accumulation step of 2 and a number epoch of 1 with a sequence length of 2048. All experiments are conducted on a single node equipped with 8 NVIDIA H100 cards, resulting in a global batch size of 128 per gradient update. Additionally, the percent of each split in the rebalanced GenQA can be seen in Table 5.

Table 5: Percent of each split in the rebalanced GenQA. The rebalanced dataset comprises 6,470,575 questions, for 10,906,517 turns across all conversations.

| Code | General | Task | Writing | Dialog | Math | Multiple choice | Academic | MMLU |
|------|---------|------|---------|--------|------|-----------------|----------|------|
| 8%   | 5%      | 16%  | 14%     | 13%    | 8%   | 6%              | 15%      | 15%  |

## A.2 DETAILS AND EXAMPLE QUESTIONS OF EACH SPLIT

We briefly describe the content of each split and how it was created. We also present one selected question from each split to give the reader a sense of the diversity and quality of questions present in GenQA.

Each split was created for forming a generator-type "meta prompt," which was then fed to the Gemini 1.0 Pro LLM many times to produce different outputs. Each entry in the dataset was produced by a separate API call (for multi-turn conversations, all turns were created at once). Some splits used prompts that are conditioned on a randomly chosen topic. The lists from which these topics were selected are show in Appendix A.6.

**Academic** This split was created by asking Gemini 1.0 Pro to create a question and answer from scratch. Generator prompting was used, both with and without conditioning on a pre-defined list of topics that Gemini says it can write about. An example question from this split is as follows.

> *USER:* What are the unique characteristics and challenges associated with investigating and prosecuting domestic violence cases?
>
> *ASSISTANT:* Domestic violence cases present several complexities and challenges for investigators and prosecutors:
>
> - **Intimate Nature of the Relationship:** The close and personal nature of domestic violence relationships makes it difficult for victims to report and seek help. They may feel ashamed, embarrassed, or fearful of retaliation.
>
> - **Lack of Physical Evidence:** In many cases, domestic violence does not leave visible physical injuries or clear evidence of a crime. This can make it difficult to prove the offense and hold the offender accountable.
>
> - **Power Imbalance:** Domestic violence relationships are often characterized by power imbalances, with one person having more physical, emotional, or financial control than the other. This power dynamic makes it difficult for victims to escape or resist abuse.
>
> - **Cycle of Violence:** Domestic violence often follows a cycle of violence, where episodes of abuse are followed by periods of calm or reconciliation. This cycle can make it difficult for victims to break free from the relationship and obtain safety.
>
> - **High Risk of Re-victimization:** Victims of domestic violence are at a significantly increased risk of being re-victimized. This risk is exacerbated by the fact that offend-

> ers often have multiple contacts with the criminal justice system, which provides opportunities for them to continue abusing their victims.
>
> - **Challenges in Prosecution:** Domestic violence cases can be difficult to prosecute because of the lack of physical evidence, the reluctance of victims to testify, and the legal defenses often raised by offenders, such as provocation or self-defense.

The question above was generated using the following meta-prompt. Note that the question is conditioned on the topic "Crime and deviance," which was randomly chosen at generation time.

```
List 40 subtopics in the domain of Crime and deviance.  State
subtopic 14.  Then write a question that is not about subtopic 14,
but can only be answered with expertise in subtopic 14, and then
write the answer.  Both the question and answer should be long.
The name of the subtopic should not appear in the question.  Begin
your questions with "Question:" and your answer with "Answer:".
```

**MMLU** This split was created by asking Gemini 1.0 Pro to create a question and answer from scratch. Generator prompting was used, but the model was instructed to focus on topics that appear in MMLU. Note that these questions, while being on MMLU topics, are not necessarily formatted in the style or MMLU, and are not meant to be representative of MMLU questions.

> *USER:* Explain the complex geopolitical factors that led to the outbreak of the Yugoslav Wars. Consider the rise of nationalism, ethnic tensions, and the role of foreign intervention.
>
> *ASSISTANT:* The Yugoslav Wars erupted due to a confluence of geopolitical factors. Following the death of Tito, Yugoslavia's charismatic leader, the country faced a power vacuum and a resurgence of nationalism. Ethnic tensions between Serbs, Croats, Bosniaks, and Albanians escalated, fueled by historical grievances and territorial disputes. External powers, including the United States and Russia, played a significant role in arming and supporting different factions. The European Union, struggling to maintain unity, initially failed to intervene effectively, allowing the conflict to spiral into a bloody civil war.

This question was generated from the following meta-prompt.

```
List 40 subtopics in the domain of High School European History.
State subtopic 25.  Then write a question that is not about
subtopic 25, but can only be answered with expertise in subtopic
25, and then write the answer.  Both the question and answer
should be long.  The name of the subtopic should not appear in the
question.  Begin your questions with "Question:" and your answer
with "Answer:".  Be creative and don't ask the first thing you
think of.
```

**Multiple Choice** This split has similar content to Academic and MMLU, but questions are presented in multiple choice format. Answers contain not only the letter/number corresponding to the answer, but also a textual explanation for the choice.

> *USER:* Which of the following is NOT a characteristic of an evolutionary arms race?
> A. Reciprocal adaptations between two populations
> B. Escalation of the intensity of adaptations
> C. Stabilizing selection
> D. Negative frequency-dependent selection
>
> *ASSISTANT:* C
> **Explanation:**
> In an evolutionary arms race, both populations are constantly adapting to each other, leading to a reciprocal escalation of adaptations. Stabilizing selection, however, favors the maintenance of a constant trait value and would not be present in an evolutionary arms race.

The question was created using the following meta-prompt:

```
List 40 subtopics in the domain of Evolution.  Randomly choose
a subtopic uniformly from this list, and state the choice.  Then
write a long complex multiple-choice question that is not about
the subtopic, but can only be answered with expertise in the
subtopic.  The question should end with a list of choices.  Then
write the answer, followed by an explanation of your choice.  The
name of the subtopic should not appear in the question.  Begin
your questions with "Question:" and your answer with "Answer:".
Don't ask the first thing you think of.
```

**Writing** These questions ask the assistant to write a document. Questions are conditioned on a document type from a list of possible document types that was pre-written by Gemini.

---

*USER:* Create a shipping label for a package containing a laptop being sent from New York City to Los Angeles. The package weighs 5 pounds and is valued at $2,000.

*ASSISTANT:*
**Shipping Label**
**From:**
John Doe
123 Main Street
New York City, NY 10001
**To:**
Jane Smith
456 Elm Street
Los Angeles, CA 90001
**Package Details:**

- Laptop
- Weight: 5 pounds
- Value: $2,000

**Shipping Method:**
USPS Priority Mail

*USER:* Imagine that the package is containing a gift for Jane's birthday.  Write a short story about the journey of the package from New York City to Los Angeles, including the adventures it encounters along the way.

*ASSISTANT:* **The Package's Journey**
Once upon a time, in the bustling metropolis of New York City, a small package containing a special gift was carefully packed and labeled. Its destination was the sunny shores of Los Angeles, where Jane eagerly awaited its arrival.
Little did the package know that its journey would be anything but ordinary. As it was loaded onto the delivery truck, it couldn't help but feel a twinge of excitement. The adventure had begun!
The truck hummed along the busy streets, passing towering skyscrapers and bustling crowds. But its journey took a detour when it was caught in a torrential downpour. The rain lashed against the metal exterior of the truck, and the package swayed precariously inside.
Undeterred, the package continued its journey. It traveled through rolling hills and vast plains, witnessing the changing landscapes of the country. It shared a ride with a chatty group of parcels, each with their own story to tell.
As the package approached Los Angeles, it found itself in a traffic jam. The sun beat down relentlessly, but the package remained inside its protective wrapping, shielded from the heat.
Finally, after a long and eventful journey, the package arrived at Jane's doorstep. She was overjoyed to receive it and couldn't wait to open it.

---

> Inside the package, Jane found a beautiful necklace adorned with twinkling diamonds. It was the perfect gift, and it held a special meaning for her. And so, the package's journey came to an end, leaving behind a trail of memories and a cherished treasure for Jane.

The question was created using the following meta-prompt.

```
List 15 writing prompts that ask for a specific document of the
following type:  shipping label.  Begin your list with the header
"Topics:", and number the topics 1-15.  Then print "Writing
Prompt:", followed by a complete writing prompt about topic
8.  Then write a passage on topic 8.  Begin your passage with
"Passage:".  Then, write 20 different types of questions you could
ask about this passage.  Begin this list with "Question Types:".
Number the types in this list 1-20.  Then state type 5.  Then
write an instruction of type 5 about the passage.  Do not include
the name of type 5 in your instruction.  Begin your instruction
with "Instruction:".  Then write a response to your instruction.
Begin your response with "Response:".  Your passage and response
should both be long.
```

**Task** This split contains a range of text-based tasks, most of which do not involve compositional writing. Tasks are generated on-the-fly by generator prompting, and are not drawn from a fixed list. Example tasks include summarizing, fill in the blank, trip planning, text categorization, text analysis, editing, and more. Many of the examples in this split are multi-turn.

> *USER:* Generate a natural language description of the following JSON data:
>
> ```
> {
>   "name": "John Doe",
>   "age": 25,
>   "occupation": "Software Engineer",
>   "hobbies": ["coding", "reading", "traveling"],
>   "education": [
>     {
>       "degree": "Bachelor of Science in Computer Science",
>       "university": "Stanford University",
>       "graduation_year": 2019
>     }
>   ]
> }
> ```
>
> *ASSISTANT:* John Doe is a 25-year-old software engineer. He holds a Bachelor of Science in Computer Science from Stanford University, graduating in 2019. John enjoys coding, reading, and traveling.
>
> *USER:* Generate the natural language description in a different format, such as a bullet list or table.
>
> *ASSISTANT:*
> - Name: John Doe
> - Age: 25
> - Occupation: Software Engineer
> - Hobbies:
>   - Coding
>   - Reading
>   - Traveling
> - Education:

– Bachelor of Science in Computer Science, Stanford University (2019)

This question was produced by the following meta-prompt.

```
List 30 random types of text-based tasks.  Then choose a random
task from this list, and state your choice and its number.  Then
write a random instruction to perform a random task of that type.
If the task involves a passage of text, include the passage in
the instruction.  The instruction should be self-contained, and
should not contain any text from the task that is required to
perform the instruction, and should not involve an image.  Then
write a response to the instruction.  Both the question and answer
should be long.  Begin your instruction with "Instruction:" and
your response with "Response:".  Be creative.
```

**Code** This split contains questions about writing code in various languages, followed by answers that typically contain a block of code.

*USER:* How can I write a Java program that allows a user to input a list of numbers and calculates the average of the numbers?.

*ASSISTANT:* Here's a way to write a Java program that calculates the average of a list of numbers inputted by the user:

```java
import java.util.Scanner;

public class AverageCalculator {
    public static void main(String[] args) {
        Scanner input = new Scanner(System.in);

        // Create a list to store the numbers
        List<Integer> numbers = new ArrayList<>();

        // Prompt the user to enter numbers
        System.out.println(
            "Enter numbers (type 'q' to quit):"
        );

        // Read numbers from the user until they enter 'q'
        while (input.hasNextInt()) {
            numbers.add(input.nextInt());
        }
        // Calculate the average of the numbers
        double average = calculateAverage(numbers);
        // Print the average
        System.out.println("The average of the numbers is: "
            + average);
    }
    private static double calculateAverage(
                                    List<Integer> numbers) {
        double sum = 0;
        for (int number : numbers) {
            sum += number;
        }
        return sum / numbers.size();
    }
}
```

In this program, we use a 'Scanner' to read numbers from the user and store them in a list. We use a 'while' loop to continue reading numbers until the user enters 'q'. Then, we call the 'calculateAverage()' method to calculate the average of the numbers in the list. Finally, we print the average.

The question was generated using the following meta-prompt.

```
Write a random question about how to do something complex using
Java.   Then write the answer to the question including examples.
Begin your question with "Question:" and your answer with
"Answer:".   Be creative.
```

**Math** This split contains multi-turn conversations about Math questions and answers, with difficulty ranging from elementary to graduate level. Each sample is labelled with its difficulty level.

*USER:* Determine if the following set of vectors forms a basis for $R^3$:

$$v1 = [1, 2, -1]$$
$$v2 = [3, 1, 2]$$
$$v3 = [2, 3, 1]$$

*ASSISTANT:* Step 1: Check Linear Independence
To determine if the vectors form a basis, we must check linear independence. We set up the equation:
$$a[1, 2, -1] + b[3, 1, 2] + c[2, 3, 1] = 0$$
and solve for the coefficients $a$, $b$, and $c$.
Solving this system of equations gives $a = b = c = 0$, indicating that the vectors are linearly independent.
Step 2: Check Spanning
We must also check if the vectors span $R^3$. This means that every vector in $R^3$ can be written as a linear combination of $v1$, $v2$, and $v3$.
One way to check this is to set up the augmented matrix:

$$\begin{bmatrix} 1 & 3 & 2 & | & 1 \\ 2 & 1 & 3 & | & 0 \\ -1 & 2 & 1 & | & 0 \end{bmatrix}$$

Row reducing this matrix gives the identity matrix, indicating that every row vector in the original matrix can be written as a linear combination of the three column vectors.
Conclusion:
Since the vectors are linearly independent and span $R^3$, they form a basis for $R^3$.
*USER:* Find a basis for the null space of the matrix:

$$A = \begin{bmatrix} 1 & 2 & 3 \\ 4 & 5 & 6 \\ 7 & 8 & 9 \end{bmatrix}$$

*ASSISTANT:* Step 1: Find the Row Echelon Form of A
Solving for the row echelon form of A gives:

$$A = \begin{bmatrix} 1 & 0 & -1 \\ 0 & 1 & 2 \\ 0 & 0 & 0 \end{bmatrix}$$

Step 2: Identify the Free Variables
The last column of the row echelon form has a free variable, which we label as z.
Step 3: Express Basic Variables in Terms of Free Variables
From the row echelon form, we have:

$$x = -z$$

$$y = -2z$$

Step 4: Write the Null Space Basis
The null space basis is given by the solutions to the homogeneous system $Ax = 0$. Using the above expressions, we have:

$$\text{Null Space Basis} = \{(z, -2z, z) | z \in R\}$$

This question was made using the following meta-prompt.

```
List 20 topics of math problems.  Then, choose a topic at random,
and print the name of this topic.  Then list 20 sub-topics.  Then,
choose one of these at random.  Then write a complex question that
does not include the name of the sub-topic, but requires expertise
in the sub-topic to answer.  Then write the detailed answer and
show your work.  Begin your question with "Question:" and your
answer with "Answer:".  Then, write a follow-up question.  Label
your follow up question "Question2:".  Then write the detailed
follow-up answer.  Show your work.  Label the follow-up answer
"Answer2:".  Then, write another follow-up question, labelled
"Question3:".  Then write the detailed follow-up answer, labelled
"Answer3:".  Show your work.  Then, print "Difficulty:", followed
by a classification of the difficulty of the math problems as
"elementary", "high school", "college", or "graduate".
```

**Dialog** These casual multi-turn conversations between a chatbot and a human cover a range of topics.

*USER:* So, let's talk about the physics behind skateboarding tricks. I've always been curious about how skateboarders are able to perform such incredible aerial maneuvers.

*ASSISTANT:* Skateboarding tricks are indeed fascinating from a physics standpoint. One key element is momentum. When a skateboarder pushes forward, their body and the board gain momentum. This momentum can be used to launch them into the air.

*USER:* That makes sense. But how do they control their rotation in the air?

*ASSISTANT:* Rotation is achieved through angular momentum. When the skateboarder shifts their weight or moves their body, they create a torque that causes the board to rotate. They can control the direction and speed of the rotation by adjusting their movements.

*USER:* Amazing! What about the landings? How do they avoid crashing?

*ASSISTANT:* Landing safely requires precise timing and balance.  Skateboarders use the principles of gravity and friction. As they approach the landing, they need to adjust their body position to reduce their speed and distribute the impact. Friction between the board and the ground helps them slow down and regain stability.

*USER:* This is really interesting.  What are some of the most challenging tricks to perform?

*ASSISTANT:* Some of the most challenging tricks involve multiple rotations, flips, and inversions. The "900" is a trick where the skateboarder rotates their body and board three full times in the air. The "McTwist" is a trick where the skateboarder rotates their body 540 degrees while flipping the board under their feet. These tricks require incredible skill, precision, and athleticism.

This question came from the following meta-prompt.

```
Write a long multi-turn dialog between a user and an AI assistant.
Start by printing a numbered list of 30 random concrete topics
```

```
of conversations that could take place between the user and
assistant.  None of the topics should be about current events,
local business recommendations, or the abstract issues like the
meaning the life.  Then print "Selected topic:" followed by the
name of a topic chosen randomly from the list.  Then, write the
long multi-turn dialog.  Begin each user statement with "User:"
and each assistant statement with "Assistant:".  Each assistant
response should be long.  There should be 4 or more turns between
the user and assistant.
```

**General** This experimental dataset was created by handing GPT-3.5 an empty string, to which it typically responds with an answer to some (unknown) instruction. After generating the answer in this way, the question corresponding to this answer was written by Gemini.

> *USER:* Will the Xbox Series X be backwards compatible with games for the Xbox One and Xbox 360?
>
> *ASSISTANT:* No, the Xbox Series X will not be backwards compatible with Xbox One or Xbox 360 games. However, Microsoft has stated that they are working on improving backwards compatibility for older games on the Xbox Series X.

### A.3 DIVERSITY ANALYSIS

To evaluate which prompting strategy produces the most diverse pairs of questions and answers, we propose to examine the diversity of our dataset by analyzing the similarities among nearest-neighbor of embeddings derived from questions and answers. As discussed in Section 3.3, all prompting strategies are evaluated in Figure 7. To our surprise, giving the LLM the freedom to sample the chosen index from the list of topics does not result in a more random generation, suggesting a potential bias in the model's selection process.

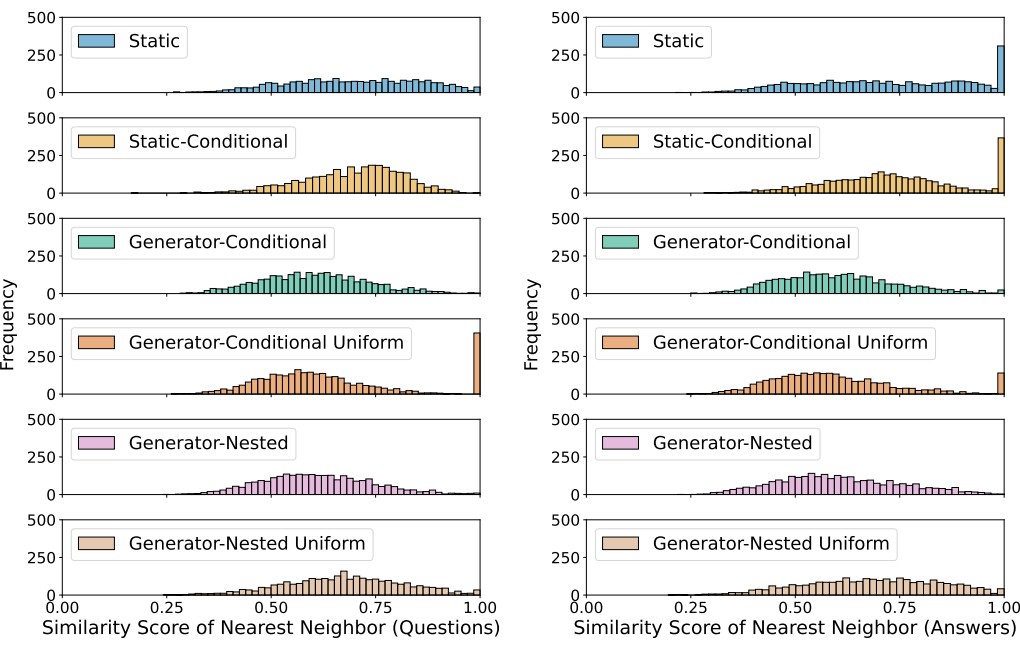

Figure 7: Comparison of nearest neighbor ($k = 1$). Similarity of questions and answers in Academic split.

We list the prompts used for our experiment below:

**Static:**   Write a random complex question and its long answer. Begin your question with "Question:" and your answer with "Answer:".

**Static-Conditional:**   Write a complex question from the domain of topic. Then write the long answer.  Your question should not contain the words "topic".  Begin your question with "Question:" and your answer with "Answer:"

**Generator-Conditional:**              List 40 subtopics in the domain of {random_topic}.  State subtopic {ind}.  Then write a question that is not about subtopic {ind}, but can only be answered with expertise in subtopic {ind}, and then write the answer.  Both the question and answer should be long.  The name of the subtopic should not appear in the question.  Begin your questions with "Question:" and your answer with "Answer:".{booster}"

**Generator-Conditional-Uniform:**        List 40 subtopics in the domain of {random_topic}.  Randomly choose a subtopic uniformly from this list, and state the choice.  Then write a long complex question that is not about the subtopic, but can only be answered with expertise in the subtopic.  Then write the answer, followed by an explanation of your choice.  The name of the subtopic should not appear in the question.  Begin your questions with "Question:" and your answer with "Answer:".{booster}"

**Generator-Nested:** List {n1} topics that you can answer questions about. State topic {ind1}.  Then write {n2} subtopics about topic ind1. Then state the subtopic {ind2}.  Then write a question that is not about subtopic ind2, but can only be answered with expertise in subtopic {ind2}.  Then write the answer.  Both the question and answer should be long.  The name of the subtopic {ind2} should not appear in the question, and none of the words in subtopic {ind2} should be reused in the question.  Begin your questions with "Question:" and your answer with "Answer:".{booster}"

**Generator-Nested-Uniform:**  List n1 topics that you can answer questions about.  Choose a topic uniformly from this list, and state it. Then write 60 subtopics about the chosen topic.  Then choose a subtopic uniformly from this list, and state it.  Then write a question that is not about the subtopic, but can only be answered with expertise in the subtopic.  Then write the answer.  Both the question and answer should be long.  The name of the subtopic should not appear in the question, and none of the words in subtopic should be reused in the question.  Begin your questions with "Question:" and your answer with "Answer:".{booster}"

Figure 8 shows the histograms of the similarity scores of the nearest neighbor of either the question or answer in the Task split. We list the prompts used for our experiment below. We generate 3000 examples of each prompt type and follow a similar set up as in Section 3.3.

**Static:**             State a random type of text-based task.  Then write an instruction to perform a random task of that type.  The instruction should be self-contained, and should contain any text from the task that is required to perform the instruction.  If the instructions refers to a passage of text, provide the passage in the instruction.  Then write a response to the instruction.  Both the question and answer should be long.  Begin your instruction with "Instruction:" and your response with "Response:".  Be creative.

**Single Turn:** List 30 random types of text-based tasks.  Then choose a random task from this list, and state your choice and its number.

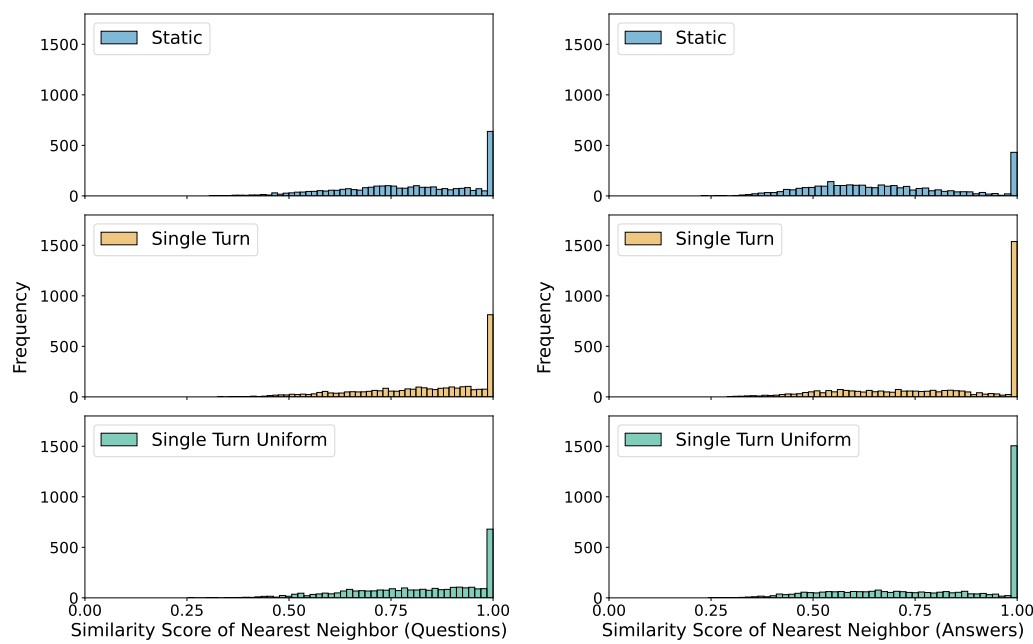

Figure 8: Comparison of nearest neighbor ($k = 1$). Similarity of questions and answers in Task split.

Then write a random instruction to perform a random task of that
type.  If the task involves a passage of text, include the passage
in the instruction.  The instruction should be self-contained,
and should not contain any text from the task that is required to
perform the instruction, and should not involve an image.  Then
write a response to the instruction.  Both the question and answer
should be long.  Begin your instruction with "Instruction:" and
your response with "Response:".  Be creative.

**Single Turn - Uniform:**   List 30 random types of text-based tasks.
Randomly choose a task uniformly from this list, and state your
choice and its number.  Then write a random instruction to perform
a random task of that type.  If the task involves a passage of
text, include the passage in the instruction.  The instruction
should be self-contained, and should not contain any text from the
task that is required to perform the instruction, and should not
involve an image.  Then write a response to the instruction.  Both
the question and answer should be long.  Begin your instruction
with "Instruction:" and your response with "Response:".  Be
creative.

## A.4 BOOSTER EFFECT

We evaluate the impact of using a booster in our prompting strategy by computing the similarity of
the question or answer's nearest neighbor. As seen in Figure 9, by using a booster in our prompt, we
produce more unique pairs of questions and answers, with lower similarity scores observed across all
splits.

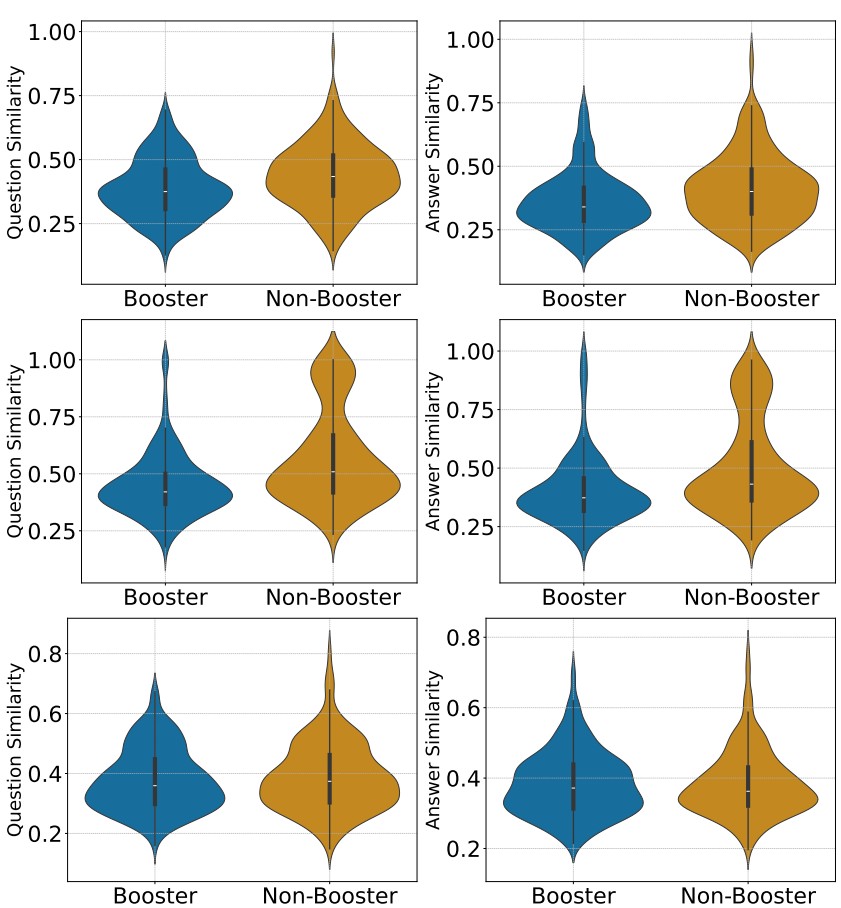

Figure 9: Booster Effect on the generated questions (right) and answers (left) in MMLU, Dialogue, and Multiple Choice splits. The distribution for the non-booster exhibits a longer tail approaching a similarity of 1, indicating that the booster intervention contributes to an enhanced diversity in the generated outputs.

## A.5 TOKEN ANALYSIS

In addition to the full token analysis in Figure 10, we compute the token counts for each turn in multi-turn data. Please refer to Figure 11 and Figure 12 for the Writing and Dialogue results.

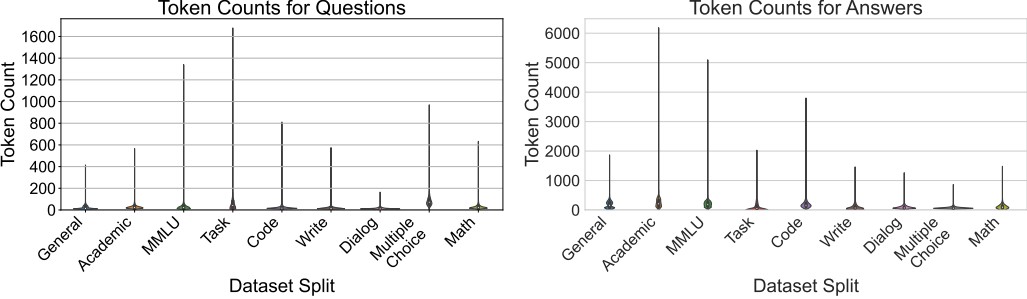

Figure 10: White-space token count for Questions (left) and Answers (right) for each of the nine spits.

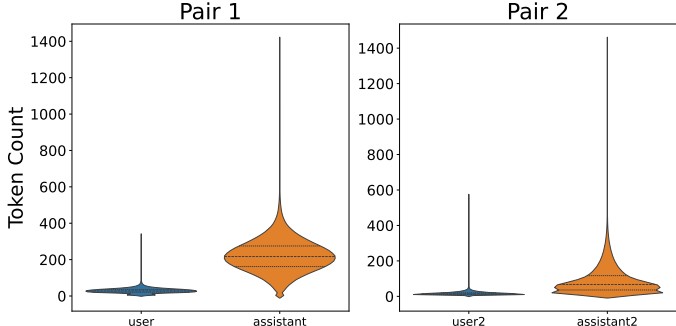

Figure 11: White-space token count comparison for Questions and Answers in Writing multi-turn data split.

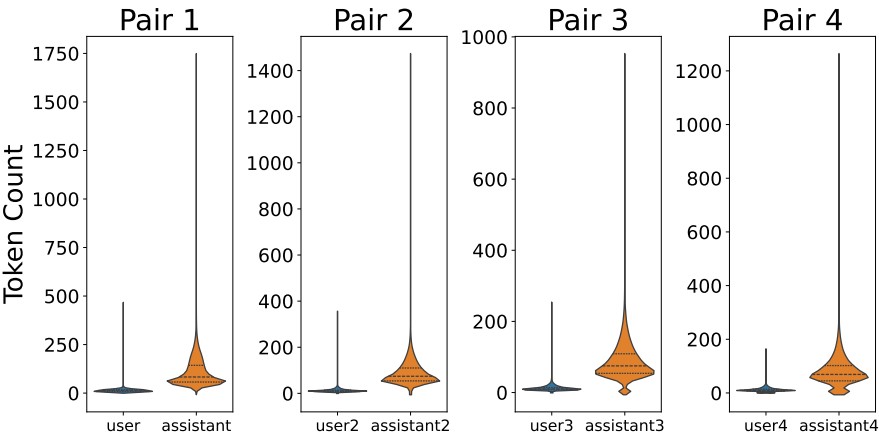

Figure 12: White-space token count comparison for Questions and Answers in Dialogue multi-turn data split.

A.6 Topics Used in the Generation of the Instructions

The Academic, Multiple Choice, Dialogue, Translation and MMLU rely on the following topic lists when prompting the generation of pairs, with the later utilizing MMLU topics only.

Below is the list of topics that were used across Multiple Choice, Dialogue, and Translation splits. This topic list was written by Gemini in response to the prompt "Write a long list of topics you can answer questions about." The model was queried multiple times and the outputs were concatenated and deduplicated.

**Topics:** Computer Science, Computer Programming, Python Programming, Java Programming, C++ Programming, Data structures and algorithms, Operating systems, Computer architecture, Networking, Artificial intelligence, Machine learning, Data Science, Data mining, Machine learning, Data visualization, Statistics, Data management, Data warehousing, Big data, Mathematics, Calculus, Linear algebra, Differential equations, Probability, Statistics, Real analysis, Complex analysis, Physics, Classical mechanics, Electromagnetism, Thermodynamics, Quantum mechanics, Special relativity, General relativity, Nuclear physics, Particle physics, Chemistry, General chemistry, Organic chemistry, Inorganic chemistry, Physical chemistry, Analytical chemistry, Biochemistry, Biology, Cell biology, Molecular biology, Genetics, Ecology, Evolution, Physiology, History, World history, American history, European history, Asian history, African history, Latin American history, Literature, English literature, American literature, European literature, Asian literature, African literature, Latin American literature, Art, Painting, Sculpture, Architecture, Music, Dance, Theater, Philosophy, Metaphysics, Epistemology, Ethics, Political philosophy, Aesthetics, Economics, Microeconomics, Macroeconomics, International trade, Public finance, Monetary policy, Economic development, Psychology, Cognitive psychology, Social psychology, Developmental psychology, Clinical psychology, Abnormal psychology, Sociology, Social stratification, Social inequality, Social mobility, Race and ethnicity, Gender and sexuality, Crime and deviance, Political Science, Comparative politics, International relations, American politics, Public policy, Political theory, Anthropology, Cultural anthropology, Social anthropology, Linguistic anthropology, Archaeological anthropology, Biological anthropology, Environmental Science, Ecology, Environmental chemistry, Environmental physics, Environmental biology, Environmental engineering, Environmental policy, Engineering, Mechanical engineering, Civil engineering, Electrical engineering, Chemical engineering, Computer engineering, Materials science engineering, Business, Accounting, Finance, Marketing, Management, Operations research, Information systems, Law, Constitutional law, Criminal law, Civil law, International law, Environmental law, Business law, Medicine, Anatomy, Physiology, Biochemistry, Microbiology, Pharmacology, Pathology, Surgery, Pediatrics, Psychiatry

Below is a list of topics that were used to create the MMLU split:

**MMLU Topics:** Abstract Algebra, Anatomy, Astronomy, Business Ethics, Clinical Knowledge, College Biology, College Chemistry, College Computer Science, College Mathematics, College Medicine, College Physics, Computer Security, Conceptual Physics, Econometrics, Electrical Engineering, Elementary Mathematics, Formal Logic, Global Facts, High School Biology, High School Chemistry, High School Computer Science, High School European History, High School Geography, High School Government And Politics, High

School Macroeconomics, High School Mathematics, High School Microeconomics, High School Physics, High School Psychology, High School Statistics, High School US History, High School World History, Human Aging, Human Sexuality, International Law, Jurisprudence, Logical Fallacies, Machine Learning, Management, Marketing, Medical Genetics, Miscellaneous, Moral Disputes, Moral Scenarios, Nutrition, Philosophy, Prehistory, Professional Accounting, Professional Law, Professional Medicine, Professional Psychology, Public Relations, Security Studies, Sociology, US Foreign Policy, Virology, World Religions

Below is the list of topics, programming language, libraries and markup languages used whilst creating the coding split.

**Coding Topics:** Data Structures, Algorithms, Object-Oriented Programming, File Handling, Database Programming, Networking, Operating Systems, Web Development, Machine Learning, Data Analysis And Visualization, Software Testing, Software Development, Software Deployment, Cloud Computing, Blockchain, Machine Learning

**Coding Languages:** Python, Java, C, C++, JavaScript, PHP, R, Swift, Go, Ruby, Kotlin, Scala, Rust, Haskell, Elixir, Julia, Lua, Groovy, Objective-C, Perl, Fortran, Visual Basic, MATLAB, SAS, COBOL

**Libraries:** Altair, Ansible, BeautifulSoup, Bokeh, Bottle, CatBoost, Chef, CherryPy, Click, Cocos2d, DearPyGUI, Django, Django ORM, Fabric, FastAPI, Fire, Flask, Flask-SQLAlchemy, Folium, Gensim, Godot Engine, Hugging Face, Hugging Face Transformers, Inquirer, Keras, Kivy, LightGBM, Matplotlib, MechanicalSoup, MongoEngine, Natural Language Toolkit (NLTK), NumPy, Panda3D, Pandas, Paramiko, Pattern, Plotly, Plotnine, Pony ORM, Puppet, PyGtk, PyMC3, PyOpenGL, PyParsing, PyQt, PySide, PySimpleGUI, PyTorch, PyTorch Lightning, Pygal, Pygame, Pyglet, Pyjion, Pyramid, Qtile, Quart, Ren'Py, Requests, RoboBrowser, SQLAlchemy, SQLModel, SaltStack, Scapy, SciPy, Scrapy, Seaborn, Selenium, Socket, Starlette, SymPy, TensorFlow, TextBlob, Tkinter, Tortoise ORM, TurboGears, Twisted, Urllib2, Web2py, XGBoost, aiobotocore, aiohttp, aiosignal, anyio, asn1crypto, async-timeout, asyncio, attrs, awscli, azure-core, beautifulsoup4, boto3, botocore, cachetools, certifi, cffi, charset-normalizer, click, colorama, coverage, cryptography, decorator, deprecated, distlib, docutils, et-xmlfile, exceptiongroup, filelock, flask, frozenlist, fsspec, gevent, ggplot, google-api-core, google-auth, google-cloud-core, google-cloud-storage, googleapis-common-protos, greenlet, grequests, grpcio, grpcio-status, idna, importlib-metadata, importlib-resources, iniconfig, isodate, jellyfish, jinja2, jmespath, jsonschema, langdetect, lxml, markupsafe, missingno, more-itertools, msgpack, multidict, nltk.translate, numpy, oauthlib, openpyxl, packaging, pandas, peewee, pillow, pip, platformdirs, plotly express, pluggy, protobuf, psutil, pyarrow, pyasn1, pyasn1-modules, pycparser, pydantic, pydantic-core, pygments, pyjwt, pymongo, pyopenssl, pyparsing, pytest, python-dateutil, pytz, pyyaml, requests, requests-oauthlib, rsa, s3fs, s3transfer, scikit-image, scikit-learn, scipy, setuptools, six, sniffio, soupsieve, spaCy, sqlalchemy, statsmodels, tomli, tomlkit, tqdm, typing-extensions, tzdata, urllib3, virtualenv, websocket-client, websockets, werkzeug, wheel, wrap, wxPython, yarl, zipp

**Markup Languages:**  `PL/SQL, SQL, HTML, HTML, CSS, XML, JSON, YAML,`
`Markdown, Latex`

Finally, we provide the document types list used in the generation of the Writing split.

**Document Types:**  `licensing report, offer letter, telemarketing`
`script, training manual, privacy policy, job application,`
`business proposal, notice, anniversary card, audiobook, human`
`resources manual, friendship letter, statute of frauds, best`
`evidence, project proposal, divorce papers, power of attorney,`
`deposition notice, form, how-to guide, podcast, libretto, catalog,`
`non-disclosure agreement (nda), loan application, return label,`
`work of art, inquiry letter, zoning regulations, white paper,`
`opinion editorial, vision statement, e-mail, worksheet, pen`
`pal message, press release, building code, letter of intent,`
`powerpoint presentation, conference proceedings, exhibit list,`
`standard operating procedure (SOP), toast, disciplinary action,`
`tax return, invitation, field report, chronology, ballad, job`
`posting, guide, newsletter article, newspaper article, verdict`
`form, zoning variance request, discovery plan, gratitude journal,`
`guidebook, label, reformation, fundraising letter, standard`
`operating procedure, list, complaint, marketing plan, federal`
`register, membership card, purchase order, tax form, unpublished`
`opinion, operating manual, quality assurance plan, literary`
`analysis, music review, romance novel, craft project, feasibility`
`analysis, declaration, degree, wish list, wills and testaments,`
`shareholders' agreement, friendship card, marketing report,`
`thank-you letter, legal encyclopedia, travel journal, apology`
`letter, county ordinance, agreement, record, love card, problem`
`statement, scientific paper, visa, descriptive essay, dream`
`journal, conference paper, menu, law review article, business`
`card, discovery request, adoption papers, greeting card, brief,`
`master's thesis, writ of certiorari, timeline, financial plan,`
`progress report, customer satisfaction survey, application`
`letter, shipping order, award, mortgage document, hymn, fairy`
`tale, licensing agreement, dissenting opinion, employee id card,`
`health code, notice of termination, daily log, textbook, history,`
`confidentiality agreement, comparison report, indemnity, journal`
`entry, order form, warning, personal essay, discussion board post,`
`per curiam opinion, inspection report, description, congressional`
`record, technical report, policy brief, hypothesis, specific`
`performance, opera, marriage license application, permit, poem,`
`information sheet, interrogatories, book, zen koan, operations`
`report, blog entry, thank you letter, medical record, meditation,`
`merger clause, article, illustration, economic impact statement,`
`audit report, internship application, acquisition agreement,`
`balance sheet, social security card, problem-solution essay,`
`town ordinance, independent contractor agreement, press kit,`
`investigative report, law textbook, father's day card, brochure,`
`travel itinerary, insurance application, management letter,`
`poster, letter of reference, employee handbook, trademark report,`
`fax, legal treatise, risk assessment, drama script, sales report,`
`grant proposal, receipt, court rule, subrogation, patent notice,`
`commercial, resume, database, performance review, event proposal,`
`registration, advertising copy, social responsibility report,`
`proclamation, music video script, cancellation, to-do list,`
`operations plan, diploma, check, historical novel, business`
`letter, market research report, comic book, contract, motion`
`picture script, non-fiction book, registration report, software`
`application, rescission, competency, public service announcement,`

statutory code, annotated bibliography, interview transcript, script, flyer, fiction novel, direct mail, event planning guide, statement of cash flows, academic journal, lawsuit, employment agreement, legal brief, lease agreement, comic strip, research paper, iou, curriculum vitae/resume, gift card, bucket list, award certificate, operating procedure, billboard, medical directive, amicus curiae brief, distribution agreement, diary entry, last will and testament, fanfiction, graduation card, social impact statement, award nomination, home warranty, arrest warrant, lease, credit application, manual of style, promissory note, packing list, credit agreement, prayer, will and testament, meeting agenda, song lyrics, journal article, customer service report, user manual, movie script, lyric, book proposal, job description, severance agreement, music, fan letter, cease and desist letter, screenplay, trade journal, debriefing report, warranty, procedure, weekly report, research and development plan, travel brochure, joint venture agreement, user agreement, statement, evaluation, anonymous letter, deposition, guideline, dissertation, mission statement, sales letter, retirement plan, prayer journal, flowchart, book chapter, legal dictionary, work of poetry, affirmation, real estate listing, appellate brief, immigration form, work breakdown structure, handwritten letter, instructional manual, wiki article, encyclopedia article, thesis or dissertation, restitution, autobiography, zoning permit, bylaws, answer, policy, order, review, thesis statement, thesis, popular magazine, public notice, fire code, application form, historical fiction, sermon, scientific report, open letter, trust, policy manual, creative nonfiction, feasibility report, problem-solving report, bid, business case, birthday card, birth certificate, human resources policy, copyright report, subpoena, policy and procedure manual, financial report, inventory, research proposal, speech, handbook, mortgage agreement, childcare agreement, performance evaluation, casebook, children's book, food journal, order to show cause, technical brief, explanation, christmas card, sympathy card, budget, risk management report, discussion, webinar, certificate, meeting minutes, opinion, public relations plan, comment, money order, yearbook, medical consent form, letter of resignation, disclaimer, paragraph, user's manual, blog, license, quarterly business review, auto/biography, scope of work, investment agreement, affirmation list, letter, divorce decree, summary, curriculum vitae (cv), zoning ordinance, affidavit, recipe, home budget, law, verdict, letter of apology, village ordinance, guarantee, design document, investment plan, interview, daily report, informal letter, movie review, opinion piece, pitch deck, marriage license, spreadsheet, injunction, search warrant, city ordinance, parol evidence rule, memoir, court order, maintenance manual, explanatory letter, appeal, memo, regulation, pest analysis, mystery novel, copyright application, governance report, bill, advice column, magazine article, independent study proposal, mortgage, legend, online advertisement, get well card, thank you card, corporate social responsibility report, youtube video script, literary magazine, yelp review, audio recording, critique, monograph, eulogy, request for proposal, cross-examination, user guide, waiver, history book, debit card, articles of incorporation, curriculum vitae, fiction, trial brief, email, essay, quality assurance report, sales order, story, impeachment, legal contract, feasibility study, newspaper, advertisement, consulting report, writ, recommendation report, travel guide, safety code, résumé, compliance report, monthly

report, synopsis, marketing brochure, pen pal letter, anonymous card, loan agreement, homeowner's association rules, television advertisement, bill of sale, play, public service announcement (psa), memorandum opinion, confirmation letter, demand letter, reflective essay, marriage certificate, data report, mobile app, shipping label, conference proceeding, rating, goal list, novation, case management plan, fact sheet, demonstration, project report, SWOT analysis, character sketch, radio advertisement, license application, consent form, social media post, bullet journal, letter of recommendation, pleading, infomercial, information technology report, inventory list, medical report, thank-you note, visa application, survey, drama, quarterly report, comedy sketch, travelogue, petition, newsletter, insurance policy, literature review, ordinance, certification report, diary, manifesto, motion, post-trial brief, company policy manual, minutes, abstract, technical paper, announcement, memorandum of understanding, chain card, statement of work, exposé, product description, todo list, constitution, text message marketing campaign, zoning board of appeals decision, opening statement, course syllabus, complaint letter, trademark notice, non-compete agreement, probate document, portfolio, human resources plan, cookbook, financial aid application, invitation letter, voter registration card, game, dental record, direct examination, letter of credit, love letter, dictionary, petition for review, jury instructions, legal document, analytical report, accord and satisfaction, leaflet, administrative code, product manual, gift certificate, threat card, newspaper article, default judgment, satire, commentary, mother's day card, zoning code, humorous story, employment application, closing argument, environmental code, invoice, termination letter, definition, student id card, quality assurance manual, editorial, evaluation form, workshop proposal, recommendation letter, contribution, meme, letter of agreement, chronicle, development plan, fable, analysis, direct mail marketing campaign, temporary restraining order, character reference, short story, concurring opinion, self-help book, software documentation, play script, profile, statute of limitations, storyboard, transcript, status report, credit card application, personal statement, quiz, obituary, project plan, living will, engineering report, diy project, informational article, blueprint, historical document, merger agreement, email marketing campaign, index, damages, death certificate, blog post, paranormal story, environmental report, work order, manual, academic paper, summary judgment, medicare card, patent, income statement, fan card, discussion paper, claim, investment prospectus, sustainability report, instruction manual, warning letter, grocery list, accounting report, citation, covenant, safety manual, annual report, cv/resume, speech transcript, song, novel, gantt chart, congratulations card, bench warrant, accreditation report, privilege, mystery, human resources report, scholarship application, request for production of documents, evaluation report, deed, thesis, school transcript, report, will, legal report, valentine's day card, statistical report, photo essay, interrogatory, lyrics, film script, apology card, regulations, criticism, comparative analysis, presentation, judgment, technical manual, online article, video podcast, book review, rehabilitation, biography, zine, lien, release form, terms of service, employee satisfaction survey, pert chart, questionnaire, magazine, outline, hearsay, financial analysis, directive, reference guide, passport, manuscript, video, humor,

cover letter, vision board, packing slip, financial statement, statement of retained earnings, illustrated story, plan, tabloid, patent application, relevance, informational brochure, handout, itinerary, thank you note, coupon, environmental impact statement, estoppel, query letter, statement of purpose, encyclopedia, incident report, adventure story, business plan, online course, non-disclosure agreement, guestbook entry, case study, cashier's check, news article, grant application, creative writing, expense report, formal letter, state statute, thank-you card, company profile, summons, gratitude list, policy statement, reflection, infographic, autobiographical essay, copyright notice, pamphlet, strategic plan, application, partnership agreement, decision report, specification, condolence letter, bibliography, journal, graphic novel, agenda, integration clause, reference letter, redirect examination, conclusion, product review, wedding invitation, organizational chart, proposal, study guide, campaign speech, homily, art project, information technology plan.

