# OpenReview forum: "GenQA: An Instruction Dataset of LLM Generated Questions and Answers"
_ICLR.cc/2025/Conference — ICLR 2025 Conference Withdrawn Submission_

### Official Review · Reviewer_uxNa · 2024-10-30

**Soundness:** 2
**Presentation:** 2
**Contribution:** 1
**Rating:** 3
**Confidence:** 4

**Summary:**

This paper introduces a new finetuning dataset, GenQA, primarily created using Gemini Pro 1.0, and explores the use of dynamic "generator" prompts to encourage diverse output. The work demonstrates that finetuning on the GenQA dataset can provide better results compares to UltraChat and WizardLM in a selection of evaluation settings. The paper claims that the dataset, generator prompts and finetuned model are released.

**Strengths:**

The work produces a resource that will be valuable to the LLM community. I also particularly appreciate the token-for-token analysis which strengthens the paper's claims around the effectiveness of the approach.

**Weaknesses:**

Figure 1 plots MTBench score (0-10) on the same y-axis as win-rate -- I'd suggest having a separate axis that correctly represents the possible range of values. I'd also consider referring to the DeepMind Math dataset as "DeepMind Math" or "DM Math" instead of "DeepMind" (https://huggingface.co/datasets/deepmind/math_dataset).

The distinction between static and generator prompts describes two different ways of interacting with LLMs. For static prompts, the authors provide an example where the LLM is queried multiple times, independently, using the same prompt. The authors claim that the lack of output diversity resulting from this process is problematic -- but it is an expected effect of sampling multiple tiems from a distribution. For generator prompts, the authors ask the LLM to output a list of elements. This is a distinctly different mode of operation in which the output of the next list element is conditioned on all previous generations. This is standard practice for various components of synth data generation pipelines. I would suggest downweighting the claims made in this section and focusing the narrative on the more interesting parts of the paper e.g. the study of how generator prompt selection impacts various characteristics of the generated data.

The booster effects are interesting, but don't answer the immediate question which is "which booster should I use for best results?". Deeper analysis here would be insightful.

I don't understand why the Length values in Table 2 for the GenQA 135M token and 238M tokens are so different, under the assumption that they are sampled from the same distribution.

There are differences across how the datasets being compared were constructed, particularly in terms of which models were used to generate them. All experiments are also performed with a single finetuning recipe and set of hyperparameters, which may be amenable to a particular data source. I am NOT suggesting repeating the experiments with different model recipes/setups, or say replicating WizardLM with Gemini Pro 1.0 to directly compare the methodologies, however, there are various effects that can be tested for e.g. overlap between the eval sets and generated datasets, that could provide some valuable insight here.

Broken reference on line 238.

Missing background: this work presents a method for generating instrictions, primarily in question format (hence the name GenQA), using generative models. There is a substantial amount of question generation work prior to the nowadays more common reformulation of general instruction-following that uses generative models to generate questions/instructions. I feel that the introduction and background section of this paper could both be made a lot richer by incorporating this prior work (a short but not comprehensive list includes https://aclanthology.org/D17-1090/, https://arxiv.org/abs/1705.00106,https://arxiv.org/abs/1905.08949, https://arxiv.org/pdf/2104.08678, https://direct.mit.edu/tacl/article/doi/10.1162/tacl_a_00415/107615/PAQ-65-Million-Probably-Asked-Questions-and-What). In particular, the related work section also seems to suggest that language models started to be finetuned with Flan, which is not accurate.

**Questions:**

Can you elaborate further on what the impact you expect this work to have? I can see the GenQA resource itself providing value to the community, but I am struggling to identify the scientific contributions. I'd like to see more analysis and discussion around the various deeper insights.

---

> ### Author Response · Authors · 2024-11-25
> **Rebuttal by Authors**
>
> > Figure 1 plots MTBench score (0-10) on the same y-axis as win-rate -- I'd suggest having a separate axis that correctly represents the possible range of values. I'd also consider referring to the DeepMind Math dataset as "DeepMind Math" or "DM Math" instead of "DeepMind" (https://huggingface.co/datasets/deepmind/math_dataset).
>
> Thank you for your suggestion. We will revise the plot in a camera ready version of our paper.
>
> > The distinction between static and generator prompts describes two different ways of interacting with LLMs. For static prompts, the authors provide an example where the LLM is queried multiple times, independently, using the same prompt. The authors claim that the lack of output diversity resulting from this process is problematic -- but it is an expected effect of sampling multiple times from a distribution. For generator prompts, the authors ask the LLM to output a list of elements. This is a distinctly different mode of operation in which the output of the next list element is conditioned on all previous generations. This is standard practice for various components of synth data generation pipelines. I would suggest downweighting the claims made in this section and focusing the narrative on the more interesting parts of the paper e.g. the study of how generator prompt selection impacts various characteristics of the generated data.
>
> Thank you for your insightful feedback. We agree with your observation that static and generator prompts represent different interaction paradigms with LLMs. As you rightly point out, the lack of output diversity when sampling independently from the same static prompt is an expected effect, as it stems from sampling multiple times from the same distribution.
>
> In our paper, the static prompts served as the motivation for developing a method to generate more diverse outputs, which led to the introduction of generator prompts. By conditioning the output of each element on previous generations, we aimed to overcome the limitations of static prompts and achieve greater diversity and coherence in the generated data. This distinction between the two types of prompts is central to our study.
>
> Taking your suggestion into account, we will downplay the claims related to the lack of diversity in static prompts and shift the narrative towards the more interesting aspect of our work: how generator prompt selection impacts various characteristics of the generated data. We believe this will help sharpen the focus on the innovative contributions of our study
>
> > The booster effects are interesting, but don't answer the immediate question which is "which booster should I use for best results?". Deeper analysis here would be insightful.
>
> In our current work, we chose to perform a more generalized analysis of the boosters to understand their overall impact across various tasks and settings. This approach allowed us to explore how boosters behave in broader contexts, helping identify general trends and patterns in performance. However, we did observe that the "be creative" prompt consistently produced the most unique samples. This prompt was particularly effective at generating diversity in the outputs. To address your comment directly, we plan to expand the analysis to provide a more detailed comparison of booster effectiveness, focusing specifically on identifying which booster yields the best results for specific types of tasks and dataset splits.
>
> >I don't understand why the Length values in Table 2 for the GenQA 135M token and 238M tokens are so different, under the assumption that they are sampled from the same distribution.
>
> Thank you for pointing out this issue. The samples are drawn from different task splits, such as math or multiple-choice tasks, which generally have shorter lengths compared to writing tasks. Consequently, the length distribution varies across the tasks. This is why we believe that the differing number of tokens may contribute to the observed variations in lengths.
>
> >There are differences across how the datasets being compared were constructed, particularly in terms of which models were used to generate them...
>
> We acknowledge that it is inherently challenging to fairly compare datasets generated with different models.
>
> >Missing background...
>
> Thank you for your suggestion, we will will include them in a camera ready version of our paper.

---

> > ### Comment · Reviewer_uxNa · 2024-11-26
> >
> > I'd like to thank the authors for their detailed response and for committing to improve the work. I will maintain my score at this stage but I believe this work has potential and would suggest that another iteration could make it stronger and substantially increase it's potential impact.

---

### Official Review · Reviewer_wTog · 2024-11-03

**Soundness:** 4
**Presentation:** 3
**Contribution:** 3
**Rating:** 6
**Confidence:** 4

**Summary:**

Summary:

The paper GenQA introduces a method for generating large-scale instruction datasets using minimal human input. By employing generator prompts, the authors autonomously generate millions of instruction-response pairs across diverse subjects. This approach minimizes reliance on human curation and bridges the gap between the scale of academic and industry datasets, producing an industrial-scale dataset intended for finetuning large language models.

Contributions:
1. Generator Prompting Techniques: The paper introduces innovative generator prompt strategies, such as generator-conditional and generator-nested prompts, which enhance data diversity by increasing randomness and avoiding repetition. These methods are shown to outperform traditional static prompts, yielding a high-coverage dataset across various question types and complexity levels.
2. Performance on Benchmark Tasks: Models finetuned on GenQA, including subsets for specific tasks, achieve competitive performance on instruction-following and knowledge-based benchmarks, such as AlpacaEval, MT-Bench, and reasoning tasks in math and general knowledge. This demonstrates that models trained on GenQA can generalize effectively to complex tasks and remain comparable to those trained on industry-standard datasets.
3. Public Release of Dataset and Methods: The authors release not only the full GenQA dataset but also the meta-prompts used to generate each data split, allowing for reproducibility and further exploration. This openness supports ongoing research on automated data generation techniques and the development of scalable datasets for finetuning large models.
4. Detailed Diversity Analysis: The paper includes an extensive quantitative evaluation of diversity across GenQA and other datasets like WizardLM and UltraChat. Using similarity scores to measure uniqueness, the authors demonstrate GenQA’s high diversity, which rivals or exceeds that of traditional, manually curated datasets.

**Strengths:**

Originality: The paper presents an innovative approach to instruction dataset creation, leveraging generator prompts, specifically the nested and conditional variants, to produce a vast dataset autonomously, without relying on extensive human oversight or labor. Additionally, by applying their methodology to generate over 10 million instructions across multiple domains, the paper highlights an impactful application of prompt engineering in scalable data generation.

Quality: The paper is grounded in rigorous empirical work, as demonstrated through its extensive evaluation of data diversity and benchmark performance. The authors also provide detailed ablations and diversity analyses that strengthen the argument for GenQA's efficacy. Furthermore, the paper thoroughly documents prompt configurations, token counts, and finetuning procedures, enhancing the reproducibility and robustness of the results.

Clarity: The paper is well-structured and clearly conveys its objectives, methodology, and findings. The description of generator prompt types is accessible and detailed, allowing readers to understand the differences in prompt configurations and their impact on data diversity. The inclusion of charts enhances readability, and the breakdown of the dataset into distinct splits provides a clear understanding of how GenQA was designed to cover a wide range of knowledge domains and instruction formats.

Significance: GenQA addresses a significant challenge in LLM training: the high cost and labor-intensive process of creating large-scale instruction datasets. By demonstrating that autonomously generated datasets can rival or exceed the performance of traditional, manually curated datasets, GenQA has the potential to reshape data generation practices in research and industry. The ability to create high-quality datasets with minimal human intervention democratizes access to LLM finetuning resources, making industrial-scale finetuning feasible in academic and open-source settings. Additionally, the public release of GenQA and its generation techniques paves the way for further research on scalable dataset generation and prompt diversity in LLM training.

**Weaknesses:**

Dependency on Generator Model Quality and Accessibility:
The quality of GenQA is highly dependent on the capabilities of the generator model, which raises potential limitations in both reproducibility and generalizability.

Lack of Analysis on Why "Bigger" Improves Only Certain Task Performance:
While the paper observes that training on the entire GenQA dataset yields improvements on instruction-following evaluations and Leaderboard tasks, it does not explore why this increased performance is limited to these specific areas. Without an in-depth analysis, it remains unclear whether the performance boost is due to the dataset’s sheer size, the diversity of instructions, or other factors like the nature of the tasks in these benchmarks.

Random Subset Selection Instead of Maintaining Split Ratios: The paper uses a random subset of GenQA to perform token-for-token comparisons with other datasets. However, this approach does not preserve the original ratios of different task types within GenQA, potentially leading to an unbalanced subset that does not fully represent the diversity of the full dataset. This could affect the validity of the comparison, as the subset may underperform or overperform on certain benchmarks depending on the distribution of task types.

**Questions:**

Suggestion about Generator Model: The authors could conduct additional tests using other available generator models, like GPT-3.5 or other open-source models, and report on the quality of data produced. This would also provide insights into the extent to which the method is generalizable across different base models, potentially increasing the impact of GenQA.

Why "Bigger" Improves Only Certain Task Performance: To provide clearer insights, the paper could benefit from an ablation study that varies dataset size and analyzes performance on different task categories, such as reasoning or domain-specific knowledge tasks. Additionally, examining the dataset’s effectiveness on a broader range of benchmarks or task-specific evaluations would clarify whether "bigger is better" applies universally or only to certain types of instruction-tuning tasks. This analysis would make the findings around dataset size more robust and actionable for future dataset design.

Suggestion about subset selection: To ensure a fairer and more representative comparison, the authors could create the GenQA subset by maintaining the same ratio of task types as in the full GenQA. This approach would better reflect the dataset’s diversity and allow for more accurate conclusions regarding the effect of dataset size on performance. This adjustment would make the subset more comparable to the full GenQA, strengthening the token-for-token analysis and ensuring it more accurately reflects the dataset’s overall characteristics.

---

> ### Author Response · Authors · 2024-11-26
>
> > Dependency on Generator Model Quality and Accessibility: The quality of GenQA is highly dependent on the capabilities of the generator model, which raises potential limitations in both reproducibility and generalizability.
>
> We recognize that relying on a single model, such as Gemini 1.0 Pro, introduces potential limitations in reproducibility, especially when other models with different architectures, training data, or fine-tuning strategies are used. In such cases, results may vary based on the particular model’s strengths and biases, affecting the consistency of outcomes across different experiments or deployments.
>
> While the current study relies on a specific generator model, we believe that this work serves as a proof of concept for the broader applicability of the GenQA framework. The financial and computational costs associated with running multiple models at scale would have added significant complexity to the project. Despite this, we see this as a crucial direction for future research, and we hope to explore it further when resources allow for a broader evaluation across different generator models.
>
>
> > Lack of Analysis on Why "Bigger" Improves Only Certain Task Performance: While the paper observes that training on the entire GenQA dataset yields improvements on instruction-following evaluations and Leaderboard tasks, it does not explore why this increased performance is limited to these specific areas. Without an in-depth analysis, it remains unclear whether the performance boost is due to the dataset’s sheer size, the diversity of instructions, or other factors like the nature of the tasks in these benchmarks.
>
>
> We appreciate your question. It's still not fully understood why larger and more diverse training datasets  lead to better model performance, even in cases like Llama 3 where explicit analysis is lacking. The relationship between dataset scale and model quality, often referred to as data scaling laws, is a complex topic that we're actively exploring in our current research.
>
>
> > Random Subset Selection Instead of Maintaining Split Ratios: The paper uses a random subset of GenQA to perform token-for-token comparisons with other datasets. However, this approach does not preserve the original ratios of different task types within GenQA, potentially leading to an unbalanced subset that does not fully represent the diversity of the full dataset. This could affect the validity of the comparison, as the subset may underperform or overperform on certain benchmarks depending on the distribution of task types.
>
>
> To improve the balance of the original GenQA dataset, we adjusted the distribution of data across its various splits. When randomly selecting over 100 million tokens, we aimed to maintain the original dataset's split ratios as closely as possible. This large sample size suggests that the new dataset's ratios will be close to the original ones.

---

> > ### Comment · Reviewer_wTog · 2024-11-29
> > **Clarifications on Data Balancing, Sampling Strategies, and Dataset Scale**
> >
> > Thank you for your detailed response regarding the data balancing and sampling strategy. I appreciate the effort to maintain the original dataset's split ratios during random subset selection. However, I would like to seek clarification on a few remaining points:
> >
> > 1. Could you provide more details on how the adjustment of data distribution across splits was achieved during preprocessing? Specifically, what steps or techniques were used to ensure balance?
> >
> > 2. In the training phase, you mentioned employing adjusted sampling ratios to up-weight smaller splits. For the random subset used in this study, was the up-weighting strategy still necessary, or did the subset's construction mitigate the need for it?
> >
> > 3. Lastly, could you specify the total number of tokens in the original GenQA dataset and in the randomly selected subset? This information would help evaluate the representativeness of the subset and the potential impact of random selection on the results.
> >
> > Thank you for addressing these points. Your clarification will help ensure a comprehensive understanding of the dataset's role in your findings.

---

### Official Review · Reviewer_4TQf · 2024-11-04

**Soundness:** 3
**Presentation:** 3
**Contribution:** 2
**Rating:** 3
**Confidence:** 3

**Summary:**

This paper describes GenQA, a large-scale instruction dataset generated from several prompts. Researchers face a challenge: existing publicly available fine-tuned datasets are small and cannot meet the demands of industrial-scale model training. To address this problem, the authors propose an automated data generation method that utilizes LLMs to generate diverse instruction examples from a single prompt. These examples cover many subjects, from simple complementary tasks to complex multi-round conversations.

**Strengths:**

1. Novelty: This paper presents a method for generating large-scale instruction datasets from single prompt, which is essential for research requiring large-scale datasets for model fine-tuning. The creation of GenQA datasets may profoundly impact the field of NLP, especially on fine-tuning LLMs. The authors detail the generator prompts strategy, a novel approach for improving the efficiency of extracting diverse questions from LLMs. This approach may also inspire the creation of other datasets.

2. Writing: This paper is well structured and logical, with clear diagrams and tables that help the reader to understand.

**Weaknesses:**

1. (Main) Experiments with different models and scales: LIMA [1] points out that a small amount of data can motivate the capabilities of the larger scale model (70B), and the authors could have conducted further comparisons at more model structures, such as Qwen2-7B. Or comparisons can be made at larger scales with small amounts of data regarding the quality of the generated data, which can strengthen the contribution of this paper.

2. Bias caused by prompt engineering: the Gemini 1.0 Pro LLM, used in the paper, may produce bias and cost constraints, and the situation in more varied LLMs, such as open source models and ChatGPT, remains unclear.

3. Cost issues are not mentioned or discussed. Specific cost analysis may need to be covered, such as balancing the consumption in the open-source model with the particular API cost and comparing the resource consumption with the existing programs.

[1] LIMA: Less Is More for Alignment. NeurIPS 2023.

**Questions:**

Refer to the above.

**Details Of Ethics Concerns:**

This work involves data generation by LLM that may have potential bias issues. However, the authors have not discussed how to alleviate potential concerns.

---

> ### Author Response · Authors · 2024-11-25
> **Rebuttal by Authors**
>
> > (Main) Experiments with different models and scales: LIMA [1] points out that a small amount of data can motivate the capabilities of the larger scale model (70B), and the authors could have conducted further comparisons at more model structures, such as Qwen2-7B. Or comparisons can be made at larger scales with small amounts of data regarding the quality of the generated data, which can strengthen the contribution of this paper.
>
> We appreciate the suggestion to explore additional models and scales. Our work primarily focused on evaluating the effectiveness of the dataset and methodology with a high-capability model.
>
> We chose to perform experiments on Llama-3-8B as it is the most recent version of one of the most popular base models for instruction finetuning available to the open source community. While we would have liked to perform tuning experiments on larger models such as the Llama 3.1 70B parameter variant, our academic budget combined with the size of the dataset itself forces us to leave these experiments to labs with larger compute resources.
>
> Future work in investigating other models and exploring scaling laws with smaller datasets could further highlight the generalizability of our approach across architectures and scales.
>
> > Bias caused by prompt engineering: the Gemini 1.0 Pro LLM, used in the paper, may produce bias and cost constraints, and the situation in more varied LLMs, such as open source models and ChatGPT, remains unclear.
>
> We agree that the specific prompt designs employed may have contributed to bias in the generated results. While we made efforts to minimize such bias by using neutral and balanced phrasing, we recognize that no prompt is entirely free from influencing the model’s outputs in specific directions.
>
> As noted, other datasets in the field are often constructed using a single API model, which can limit the variability and potential for bias in prompting strategies. While we focused on a generalized analysis using the Gemini 1.0 Pro LLM, we agree that it would be an insightful direction for future work to explore how well the generator prompting strategy performs across multiple APIs, including other commercial and open-source models like ChatGPT or alternative open-source models. However, due to cost constraints, this comparison was unfortunately out of scope for our current study. The computational costs associated with testing multiple API models at scale would have required significant resources, making it challenging to conduct a thorough comparison across several models. Nevertheless, we see this as an important avenue for future research, and we hope to address it in subsequent studies when resources allow for a broader evaluation of generator prompting strategies across various models
>
> > Cost issues are not mentioned or discussed. Specific cost analysis may need to be covered, such as balancing the consumption in the open-source model with the particular API cost and comparing the resource consumption with the existing programs.
>
> Thank you for your feedback. We appreciate your suggestion regarding the inclusion of a cost analysis, and we agree that it is crucial to consider both the economic and resource implications when using open-source models and APIs for large-scale tasks such as data generation. We have conducted a cost analysis for generating the entire dataset of 10 million samples.
> Total Expected Cost: **$2114.59**:
>
> This cost includes both input processing and output generation based on the average sample lengths (per task) and token pricing:
> - **Input Cost:** The cost to process 10 million samples is estimated at **$0.075 per million tokens**.
> - **Output Cost:** The cost to generate the corresponding output is estimated at **$0.30 per million tokens**.
>
> The average length of a sample varies across different splits, and the cost calculation considers both the input and output token consumption. If we were to use a commercial API (based on typical token pricing), the costs for the same dataset exceeds **$10,000**. Specifically, using GPT4 to produce this dataset would average a total cost of **$507,502.00**.

---

### Official Review · Reviewer_e9sa · 2024-11-05

**Soundness:** 2
**Presentation:** 2
**Contribution:** 2
**Rating:** 3
**Confidence:** 4

**Summary:**

This paper introduces a method for generating large-scale QA pairs for instruction tuning by prompting Gemini. The method instructs Gemini to list topics and sub-topics within a given domain and select one of them (randomly or according to a given index) and move forward to create a question-answer pair. This process introduces diversity in the QA pairs due to the randomness in listing and selecting topics (or sub-topics).

Fine-tuning a Llama-3 8B base model on large scale GenQA dataset with these QA pairs shows improved performance on knowledge-intensive tasks.

**Strengths:**

- the proposed method is highly scalable

- target domain of generated instructions can be controlled

**Weaknesses:**

- Some aspects of the method remain unclear. In Section 3.2, five different prompts are introduced, but it is not specified which ones are actually used in the experiments. I assume a combination of “Generator-Conditional,” “Generator-Nested,” and “Generator-Uniform” is applied. How are their proportions determined across different domains?

- The generated QA pairs are produced at a large scale. However, the results in Tables 3 and 4 raise potential concerns regarding their quality.

- A total of 513,483 coding QA pairs are generated (see Table 1), yet no evaluations are conducted on the coding domain in the experiments.

- The proposed method can be considered a black-box distillation approach for Gemini. Therefore, the results for Gemini should be included in Table 3 and Table 4 as an upper bound reference.

- Although 2.4 million examples are generated based on topics from the MMLU benchmark, the MMLU results in Table 2 show a decrease compared to the Llama3 base.

- Why are the MATH and GSM scores for MathInstruct so low in Table 4? According to https://arxiv.org/pdf/2309.05653 based on Llama2 7B, MathInstruct achieves a score of 31.5 on MATH. I expect with llama3, the results should be higher.


**Missing citation:**
Synthetic Data (Almost) from Scratch: Generalized Instruction Tuning for Language Models

**Questions:**

- How do you ensure that the generated questions in the QA pairs are reasonable and answerable? For instance, is there a possibility that some generated math questions may be unsolvable?

- If incorrect answers are generated, does your method filter out these examples?

---

> ### Author Response · Authors · 2024-11-25
> **Rebuttal by Authors**
>
> > Some aspects of the method remain unclear. In Section 3.2, five different prompts are introduced, but it is not specified which ones are actually used in the experiments. I assume a combination of “Generator-Conditional,” “Generator-Nested,” and “Generator-Uniform” is applied. How are their proportions determined across different domains?
>
> Thank you for your feedback. The experiment discussed in section 3.3 was conducted specifically to determine which prompt performed best for the given task. Based on those results, the selected prompt(s) were used in the generation and experimental parts. Additionally, all five prompts introduced are included in the final dataset to enable more systematic and controlled experiments across different domains.
>
> >The generated QA pairs are produced at a large scale. However, the results in Tables 3 and 4 raise potential concerns regarding their quality.
> > A total of 513,483 coding QA pairs are generated (see Table 1), yet no evaluations are conducted on the coding domain in the experiments.
> > The proposed method can be considered a black-box distillation approach for Gemini. Therefore, the results for Gemini should be included in Table 3 and Table 4 as an upper bound reference.
> > Although 2.4 million examples are generated based on topics from the MMLU benchmark, the MMLU results in Table 2 show a decrease compared to the Llama3 base.
>
> We agree that the slightly reduced performance on some of the knowledge intensive benchmarks suggests that further research is needed to fully understand the best practices in finetuning on multi-million sample instruction datasets. In light of recent industry successes in finetuning at this scale [1] enabling this type of research in an open manner was one of the motivations of our work. That said, we note that the even the official Llama3-8B-Instruct model trained on a private multi-million example instruction and preference tuning dataset also performs slightly worse than its Llama-3-8B base model counterpart on some of the benchmarks in Table 3. The main point of these results is to show that while tuning on our dataset yields marked improvements in other evaluations (Table 2 and 4) it does not compromise knowledge intensive benchmark performance in any systematic way. Finally, we note that the GenQA dataset comprises multiple separate splits that are meant to be remixed and reused for different purposes. For this reason, it is desirable to have these splits be larger than necessary so that they can be upweighted.
>
> > Why are the MATH and GSM scores for MathInstruct so low in Table 4?
>
> We think this difference arises from the evaluation methods. According to the paper [https://arxiv.org/pdf/2309.05653], they use CoT prompting or PoT prompting, and evaluate both 8-shot in-context-learning and zero-shot setups to report the highest score. In contrast, our paper evaluates using only 5 shots.

---

### Note · Authors · 2024-12-15

I have read and agree with the venue's withdrawal policy on behalf of myself and my co-authors.